# G-Sim: Generative Simulations with Large Language Models and Gradient-Free Calibration

**Samuel Holt** [* 1]  **Max Ruiz Luyten** [* 1]  **Antonin Berthon** [1]  **Mihaela van der Schaar** [1]

## Abstract

Constructing robust simulators is essential for asking "what if?" questions and guiding policy in critical domains like healthcare and logistics. However, existing methods often struggle, either failing to generalize beyond historical data or, when using Large Language Models (LLMs), suffering from inaccuracies and poor empirical alignment. We introduce **G-Sim**, a hybrid framework that automates simulator construction by synergizing LLM-driven structural design with rigorous empirical calibration. G-Sim employs an LLM in an iterative loop to propose and refine a simulator's core components and causal relationships, guided by domain knowledge. This structure is then grounded in reality by estimating its parameters using flexible calibration techniques. Specifically, G-Sim can leverage methods that are both **likelihood-free** and **gradient-free** with respect to the simulator, such as **gradient-free optimization** for direct parameter estimation or **simulation-based inference** for obtaining a posterior distribution over parameters. This allows it to handle non-differentiable and stochastic simulators. By integrating domain priors with empirical evidence, G-Sim produces reliable, causally-informed simulators, mitigating data-inefficiency and enabling robust system-level interventions for complex decision-making.

## 1. Introduction

Simulations are essential for testing decisions, developing policies, and optimizing resource allocation in domains ranging from healthcare to supply-chain management (Law & Kelton, 2000; Banks, 1998; Banks et al., 2010). A well-constructed simulator allows asking "*what if ...?*" and evaluating interventions or stress tests without bearing the risk, cost, or challenges of real-world experiments (Oliver, 2023).

Yet, manually building these simulations is often a time-consuming, resource-intensive process demanding substantial expert knowledge. The rise of Large Language Models (LLMs), with their vast general knowledge and reasoning capabilities (Bommasani et al., 2021; Chen et al., 2021), coupled with the increasing availability of observational data—albeit often fragmented—presents a compelling opportunity to automate simulator construction. Yet, despite this potential, a comprehensive framework to fully realize it has remained elusive.

Existing automated approaches, often termed "world models" (Ha & Schmidhuber, 2018; Tang et al., 2024), typically focus on model-based reinforcement learning. They estimate environment transitions and rewards to improve planning (Luo et al., 2024), primarily answering questions such as, *"What if the actor follows policy [X]?"*.

**Toward General-Purpose, Intervenable Simulators.** In contrast, a truly general-purpose simulator must enable deeper investigations into the environment's dynamics, addressing questions such as: *"What if this underlying component changes?"* or *"Is the system robust under this structural stress-test?"*. Answering these requires a new class of simulators supporting flexible, **(P0) System-wide Experimentation**. These simulators must integrate diverse data sources, handle uncertainty, and generalize effectively. To be effective, such simulators must possess several key properties:

**(P1) Plausible Generalization:** Align with domain insights, even out-of-distribution.
**(P2) Empirical Alignment:** Match available observational data.
**(P3) Data Form Consistency:** Preserve the nature (continuous, discrete, stochastic) of real-world components, since knowing a distribution can lead to drastically different conclusions than using just the mean, especially in high-stakes scenarios.

These properties ensure the simulator is both scientifically valid and practically useful.

---

[*]Equal contribution [1]University of Cambridge. Correspondence to: Samuel Holt <sih31@cam.ac.uk>.

*Proceedings of the $42^{nd}$ International Conference on Machine Learning*, Vancouver, Canada. PMLR 267, 2025. Copyright 2025 by the author(s).

**G-Sim: A Hybrid Approach.** To instantiate a simulator-builder meeting these needs, we introduce **G-Sim**, a framework for automatic environment generation that uniquely merges LLM-driven structural reasoning with robust, data-driven calibration (Figure 1). G-Sim operates through an iterative loop:

1. **LLM-Driven Structural Reasoning (P1, P3):** An LLM, prompted with domain knowledge, proposes and refines the simulator's structure (submodules, causal links), injecting expert priors for plausible dynamics.

2. **Flexible Parameter Calibration (P2):** We calibrate these structures against data using a choice of likelihood-free and gradient-free techniques. This includes *gradient-free optimization (GFO)* (Toklu et al., 2023) for parameter estimation or *simulation-based inference (SBI)* for principled uncertainty quantification.

3. **Iterative Refinement (P1–P3):** Diagnostics (e.g., predictive discrepancies) flag weaknesses, guiding the LLM via in-context learning to restructure and improve the model until satisfactory alignment is achieved.

This cycle yields a "refinement loop" where the simulator's structure and parameters co-evolve, ensuring it is causally plausible, empirically grounded, and addresses **(P0)**–**(P3)**.

**Contributions.** Our work makes several contributions:

- We introduce a *novel problem framing* for environment-building, centered on system-level experimentation for real-world decision-making (Section 2).
- We propose **G-Sim**, an *hybrid framework* combining LLM-guided structural search with flexible, data-driven calibration via a choice of GFO or SBI (Section 3).
- We demonstrate G-Sim's ability to achieve *plausible generalization* and support new forms of system-level analysis through experiments on three diverse environments (Section 5).

## 2. Problem Setting

We aim to build a *simulator* $\mathcal{M}$ that mirrors the evolution of a real-world system, enabling rigorous "*what if...*" experimentation and *policy*[1] analysis. Formally, this simulator should:

1. Produce trajectories in a state space $\mathcal{X}$ that corresponds directly to real-world configurations (**P3**).

2. Encode accurate transitions under both in-distribution and novel (out-of-distribution) conditions (**P1**–**P2**).

---
[1]We italicize *policy* when it refers to interventions potentially more general term than a typical RL policy (i.e. changes in the environment itself, such as changing the physical layout of an environment.).

3. Support submodule-level refinements and compositional design to facilitate targeted updates and domain adaptation (**P0**).

### 2.1. System State and Update Mechanisms

Let $\mathcal{X}$ be the (potentially high-dimensional) space of *system states* and $\mathbf{x}_t \in \mathcal{X}$ the state at time $t$. Let $\mathbf{u}_t \in \mathcal{U}$ denote exogenous controls, actions, or *policy* interventions. We define a simulator $\mathcal{M}$ by a *transition operator*

$$F : \mathcal{X} \times \mathcal{U} \times \Theta \rightarrow \mathcal{X},$$

with parameter space $\Theta$. In a discrete-time setting:

$$\mathbf{x}_{t+1} = F\big(\mathbf{x}_t, \mathbf{u}_t; \theta\big), \tag{1}$$

where $\theta \in \Theta$ encodes all parameters—both structural and numerical—specifying how the simulator evolves. Such parametric state-transition models have a long history in control theory (Åström, 1970).

**Submodule Partitioning and Composition.** Complex systems often factorize into smaller sub-processes (submodules). Concretely, let

$$\mathcal{M} = \{\, \mathcal{M}_1, \mathcal{M}_2, \ldots, \mathcal{M}_K \,\}$$

be a collection of submodules. Each $\mathcal{M}_k$ yields a local mapping

$$F^k : \mathcal{X} \times \mathcal{U} \times \Theta^k \rightarrow \mathcal{Y}^k,$$

where $\Theta^k \subset \Theta$ is the submodule's parameter subset and $\mathcal{Y}^k$ is an intermediate output space (e.g., a partial update or a rate in a Markov jump process). The global transition operator $F$ then composes these submodule outputs:

$$\mathbf{x}_{t+1} = F_0\Big(F^1(\mathbf{x}_t, \mathbf{u}_t; \theta^1), \ldots, F^K(\mathbf{x}_t, \mathbf{u}_t; \theta^K), \theta^0\Big), \tag{2}$$

where $\theta^0$ captures cross-submodule coupling (e.g., shared constraints, resource balances). Such compositional frameworks align with agent-based models (Bonabeau, 2002), system dynamics approaches, and block-structured simulations (Law & Kelton, 2000; Banks et al., 2010). They allow asynchronous or continuous-time versions by replacing (1)–(2) with differential equations or event-driven formulations.

**Structural vs. Numerical Parameters.** We partition the simulator's parameter space $\Theta$ as

$$\Theta = \Lambda \times \Omega,$$

$$\lambda \in \Lambda \text{ (structural params)}, \quad \omega \in \Omega \text{ (numerical params)}.$$

The structural part $\lambda$ indicates which submodules are active (e.g., "Does this subsystem exist?") or which causal links connect them (e.g., "Is submodule A driven by B's output?"), while the numerical part $\omega$ encodes real-valued

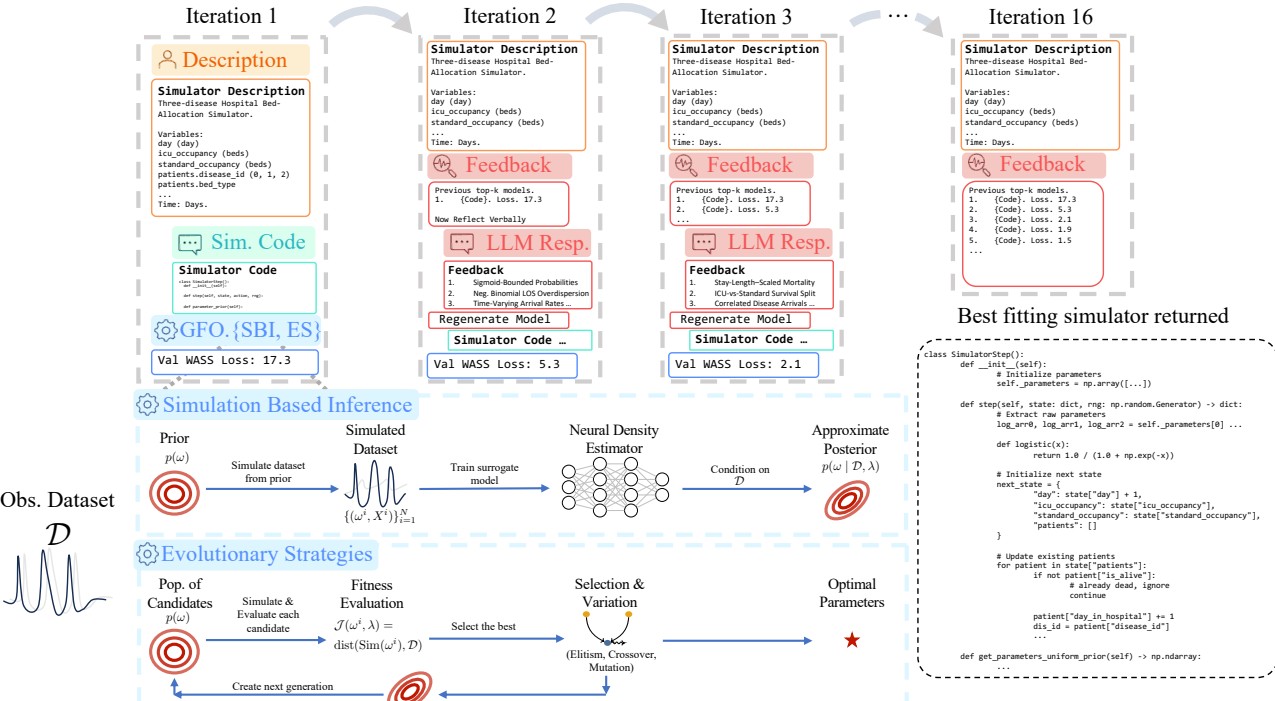

Figure 1. **Overview of the G-Sim framework for automatic simulator generation.** The process integrates LLM-driven structural design with empirical observational data ($\mathcal{D}$) in an iterative refinement loop. **(1) Propose:** An LLM first generates simulator code ($\lambda$) from a textual description. **(2) Calibrate:** This code's numerical parameters ($\omega$) are then calibrated against data using one of two parallel, likelihood-free pathways: either **Evolutionary Strategies** (a form of GFO) to find optimal parameters by minimizing a fitness function, or **Simulation-Based Inference (SBI)** to infer a full posterior distribution over parameters. **(3) Refine:** The performance of the calibrated model (e.g., validation loss) is synthesized into a natural language **feedback** summary. This summary, along with past models, guides the LLM to propose an improved structure in the next iteration. This cycle continues until performance converges, yielding a robust, empirically-grounded simulator.

or discrete parameters (e.g., rates, coefficients, or threshold levels). This factorization is particularly conducive to domain-knowledge infusion, as experts or language models can propose plausible topologies (i.e., $\lambda$) without specifying precise numerical values.

## 2.2. Queries Enabled by the Simulator

**Out-of-distribution Exploration.** From an initial condition $\mathbf{x}_0$ and a sequence of inputs $\{\mathbf{u}_t\}$ potentially outside historical distributions, the simulator generates:

$$\{\mathbf{x}_1, \ldots, \mathbf{x}_T\} = \left\{ F(\mathbf{x}_0, \mathbf{u}_0; \theta), F(\mathbf{x}_1, \mathbf{u}_1; \theta), \ldots \right\}.$$

Robust extrapolation to these unseen regimes is critical for stress-testing and scenario planning (Oliver, 2023; Rosenberger, 1993).

**Submodule-level Interventions.** More fundamentally, since $\theta = (\lambda, \omega)$ partitions structural and numerical parameters, a user may choose to modify or replace a subset of submodules $\theta^k$. Such modular updates are well-matched

to object-oriented simulation toolkits (Shewchuk & Chang, 1991), and are also valuable for stress-testing, scenario planning, and continous learning.

**Policy Analysis.** As done in previous work (Tang et al., 2024), the simulator can also serve as a decision-support tool by evaluating policy effects in a controlled environment, which is central to model-based RL (Sutton & Barto, 2018) and offline policy evaluation (Uehara et al., 2022).

## 2.3. Data and Domain Knowledge

**Observational Data with Partial Overlaps.** Real systems often produce fragmented data from multiple sources. Let $\mathcal{D} = \left\{ \mathcal{D}^{(1)}, \ldots, \mathcal{D}^{(L)} \right\}$, where each dataset $\mathcal{D}^{(l)}$ typically logs partial trajectories (e.g., some submodules but not others) or covers different time spans. Specific limitations include:

1. *Sparse coverage*: States or interventions of interest (e.g., extreme disruptions) rarely appear in observational data.

2. *Asynchronous logging*: Submodules might be sampled at varying rates and be of different types (discrete, continuous, stochastic, etc.).

3. *Privacy and partial observability*: Some datasets might not be paired or crucial variables (e.g., patient data) may not be directly recorded.

Purely data-driven fitting often struggles here, as the disjoint or partial data coverage renders many submodules unidentifiable. Causal inference literature (Pearl, 2009; Peters et al., 2017) demonstrates that even when data are plentiful, lacking the right structural assumptions can make generalization under interventions fundamentally ill-posed.

**Domain Knowledge.** Alongside $\mathcal{D}$, we assume access to domain knowledge $\mathcal{K}$, encompassing:

- *LLM Guidance*: Large Language Models can incorporate extensive textual corpora and suggest plausible topologies or parameter defaults. Their ability to generate semantically consistent code or functional forms has been noted in (Chen et al., 2021; Li et al., 2022).
- *Textual resources*: Manuals, guidelines, or domain-specific documentation of operational protocols.
- *Symbolic constraints or causal graphs*: Hard constraints (e.g., "throughput cannot exceed capacity") or partial causal diagrams (Spirtes et al., 2001).

The challenge is to integrate $\mathcal{K}$ with $\mathcal{D}$ in a balanced way, ensuring that submodule structures and parameters remain *consistent with known causal principles* while also aligning with empirical evidence.

### 2.4. Failure Modes of Naive Approaches

**Purely Data-Driven Fitting.** One might attempt to learn a single generative model $\hat{\mathcal{M}}$ from $\mathcal{D}$ by maximizing a likelihood or minimizing reconstruction error. However:

- *Lack of structural priors*: When coverage of certain interventions is sparse, extrapolation is not only statistically weak but can be *causally* ill-posed (Pearl, 2009; Peters et al., 2017).
- *Missed cross-submodule interactions*: Limited pairing or partial observability across submodules prevents coherent joint estimation.
- *Rigid optimization requirements*: Many generative modeling approaches require differentiability, hindering the inclusion of discrete or combinatorial elements (Salimans et al., 2017).
- *No intervention-readiness*: The black-box nature of many data-driven approaches makes it inherently hard to intervene beyond shifting their inputs (Shin et al., 2022).

**Purely LLM-Generated Simulators.** Conversely, one might rely entirely on LLMs to propose equations, code, or entire submodules from textual guidance:

- *Mismatched real-world statistics*: Without quantitative calibration, subtle parameter errors can accumulate and degrade fidelity even on in-distribution settings (Lian et al., 2024; Vafa et al., 2024).
- *Undetermined modules:* No mechanism exists to calibrate or refine key parameters that are undetermined from $\mathcal{K}$.

Neither approach alone suffices for demanding tasks such as *policy* evaluation or out-of-distribution stress testing.

### 2.5. Need for a Hybrid Framework

Given these limitations, we advocate a *hybrid* solution that integrates:

1. *LLM-driven structural proposals* ($\lambda$) to incorporate domain knowledge and causal heuristics, ensuring the simulator remains grounded in plausible mechanisms.

2. *Rigorous calibration* of numerical parameters ($\omega$) to observational data. This is achieved via techniques that are both **likelihood-free** and **gradient-free** w.r.t. the simulator, such as GFO for point estimation (Sehnke et al., 2010) or SBI for Bayesian inference (Cranmer et al., 2020).

$$\underbrace{\text{"What if?"}}_{\textbf{(P0)}} \longleftarrow \underbrace{\text{OOD gen.}}_{\textbf{(P1)}} + \underbrace{\mathcal{D} \text{ align.}}_{\textbf{(P2)}} + \underbrace{\mathcal{D} \text{ form consist.}}_{\textbf{(P3)}}$$

By balancing domain-knowledge based structure with empirical alignment, we move beyond purely data-driven or purely knowledge-based simulators to achieve robust *policy* evaluation and discovery in complex, real-world domains.

## 3. G-Sim: Hybrid Simulator Construction

We present **G-Sim**, a novel framework for *automatic simulator generation* that synergizes **LLM-driven structural reasoning** with **rigorous empirical calibration**[2]. As depicted in Figure 1, G-Sim operates through an iterative cycle, orchestrating three core phases: (1) proposing a simulator's architecture using an LLM, (2) grounding this structure in data by calibrating its parameters, and (3) refining the architecture based on diagnostic feedback. This section details each phase, highlighting how G-Sim achieves plausible generalization (**P1**), empirical alignment (**P2**), data-form consistency (**P3**), and system-wide experimentation (**P0**).

---

[2]Code is available at https://github.com/samholt/generative-simulations and we provide a broader research group code base at https://github.com/vanderschaarlab/generative-simulations

## 3.1. LLM-Driven Structural Design

**Proposing Compositional Structures.** Real-world systems often decompose into interconnected submodules, each governing specific dynamics like queueing, resource management, or disease progression (Shanthikumar & Wu, 1991; Choudhury & Basak, 2018). G-Sim leverages this by having an LLM propose a *structural configuration*, $\lambda$. This configuration specifies which *submodule templates* (e.g., an $SIR$ model (Kermack et al., 1997; Batista et al., 2020)) are active and how they are linked by *coupling rules*. This modular, block-structured approach (Section 2) facilitates interpretable and intervenable designs (Shewchuk & Chang, 1991), providing a strong inductive bias for causal plausibility (Klinger et al., 2023; Schug et al., 2024).

**Injecting Domain Knowledge via LLMs.** We employ an LLM as a generative engine to explore the space of these structural configurations. Prompted with domain knowledge $\mathcal{K}$ (textual descriptions, known constraints, see Appendix E.4), the LLM generates simulator code: $\lambda \sim p_{\text{LLM}}(\lambda \mid \mathcal{K})$. For example, given a description of hospital workflows, it might propose modules for patient arrivals, bed allocation, and discharge, linking them appropriately. This process injects domain-level causal hypotheses and expert heuristics (Pearl, 2009) directly into the simulator's structure, fostering plausible generalization (**P1**) and ensuring consistency with real-world mechanisms (**P3**). We assume that the LLM, guided by $\mathcal{K}$ and iterative feedback (see §3.3), can explore a sufficiently rich space of structures, including those closely approximating the true underlying system.

## 3.2. Empirical Grounding via Likelihood-Free Calibration

While the LLM defines the simulator's structure ($\lambda$), its numerical parameters ($\omega$) must be aligned with empirical data ($\mathcal{D}$). LLMs alone are often unreliable for precise quantitative estimation (Vafa et al., 2024). G-Sim addresses this by treating the simulator as a black box and offering a choice between two powerful calibration approaches that are both **gradient-free** and **likelihood-free**. This provides maximum flexibility, accommodating the non-differentiable, stochastic, and discrete components common in real-world systems.

**Pathway 1: Parameter Estimation with Gradient-Free Optimization (GFO).** The first option is to use GFO to find a single best-fit set of parameters. We use evolutionary strategies (ES), implemented via EvoTorch (Toklu et al., 2023), to find a point estimate $\omega^*$ that minimizes a *fitness function*, $\mathcal{J}(\omega, \lambda)$. This function measures the discrepancy (e.g., MSE or MMD) between simulated trajectories and observed data $\mathcal{D}$. By not requiring gradients of the simulator,

GFO excels at navigating the complex and often non-smooth loss landscapes of realistic simulators. Full details are in Appendix E.3.1.

**Pathway 2: Bayesian Inference with Simulation-Based Inference (SBI).** Alternatively, when quantifying parameter uncertainty is crucial, the user can choose SBI (Cranmer et al., 2020). SBI is a principled Bayesian framework for problems with intractable likelihoods but accessible simulators. We primarily use Neural Posterior Estimation (NPE), where a neural network (see Appendix E.3.2) is trained to approximate the posterior distribution $p(\omega \mid \mathcal{D}, \lambda)$. From this learned posterior, a point estimate for the parameters (e.g., the posterior mean or mode) is selected to instantiate the final simulator. This approach provides not just a single set of parameters but also a full characterization of their uncertainty, which is vital for assessing model confidence.

### 3.2.1. A KEY CAVEAT WHEN USING SBI

SBI's core strength is delivering principled uncertainty quantification. The learned posterior $p(\omega \mid \mathcal{D}, \lambda)$ allows for robust analysis of parameter credible intervals and correlations. However, it is crucial to acknowledge a fundamental assumption: SBI's theoretical guarantees hold when the simulator's structure ($\lambda$) is *correctly specified* (Cranmer et al., 2020). In G-Sim, we are actively *searching* for this structure. Therefore, when we perform SBI with a candidate structure $\lambda^{(g)}$, the resulting posterior $p(\omega \mid \mathcal{D}, \lambda^{(g)})$ is conditioned on a potentially misspecified model. While this posterior is invaluable for calibrating the *given* structure, its uncertainty estimates do not capture the *structural uncertainty* of the model search itself. This highlights the synergistic, yet distinct, roles of LLM-driven structural search and SBI-based parameter inference within G-Sim (Appendix B.4).

## 3.3. Diagnostics-Driven Iterative Refinement

A proposed structure, even when calibrated, might still exhibit inaccuracies or miss crucial dynamics. G-Sim addresses this via an *iterative refinement loop* that identifies weaknesses and guides the LLM toward better designs.

**Diagnostic Evaluation.** After calibration, we evaluate the current simulator ($\lambda, \omega^*$) using a diagnostic function, $\text{Diag}(\lambda, \omega^*)$. This function aggregates signals indicating mismatch, such as:

- **Predictive Discrepancy** ($\delta_{\text{predictive}}$): Metrics like Wasserstein distance or MSE comparing simulated trajectories to held-out data (Appendices E.8 and H).
- **Domain Violations** ($\delta_{\text{domain}}$): Checks for compliance with known rules (e.g., capacity limits, conservation laws) or plausibility under stress tests (Rauba et al., 2024; Li & Yuan, 2024).

The iteration loop continues for either $m$ total iterations (e.g., $m = 16$) or until Diag is below a convergence threshold $\varepsilon$ for the fitness function (see Appendix B.2).

**Textual Feedback for In-Context Learning.** When refinement is needed, G-Sim synthesizes the diagnostic findings into a *natural language summary*. For example: "*The simulator overestimates ICU occupancy during weekends and fails to capture the weekly seasonality present in the data. Consider adding a time-dependent factor to arrival or discharge modules.*" This text is fed back into the LLM's prompt, leveraging its in-context learning capabilities to guide the proposal of a revised structure, $\lambda^{(g+1)}$. This cycle of proposing, calibrating, and refining (see Algorithm 1) allows G-Sim to converge towards a simulator that is causally plausible, empirically aligned, and robust.

### 3.4. Practical Considerations: Automation, Expertise, and Prompts

While the G-Sim loop (Figure 1, Algorithm 1) is designed for a high degree of automation, practical deployment benefits from a nuanced understanding of its operation. The core iterative process runs automatically once initial domain knowledge is provided. However, human expertise can be optionally integrated. Domain experts can validate LLM-proposed structures against domain insights, interpret complex diagnostic results, or suggest specific stress tests. This "expert-in-the-loop" approach enhances trust and robustness, particularly in high-stakes applications.

Our prompt engineering strategy aims for efficiency: we use general reusable core prompts (Appendix E.4) that outline the task and code structure, supplemented by concise and environment-specific details. This requires only moderate effort, rather than extensive, custom-designed, prompt design. Detailed implementation specifics and code examples can be found in Appendix E.

## 4. Related Work

We position **G-Sim** within four major research streams—data-driven world models, foundation-model-based and LLM-coded world models, and hybrid digital twins—highlighting key limitations that G-Sim overcomes for real-world simulation-building (an extended survey is provided in Appendix A).

**Data-Driven World Models.** A large body of model-based reinforcement learning work focuses on purely data-driven approximations of environment dynamics (Ha & Schmidhuber, 2018; Hafner et al., 2019; Alonso et al., 2023; Micheli et al., 2023; Hafner et al., 2023; Ding et al., 2024; Bruce et al., 2024). While these *world models* effectively predict transitions and rewards in-distribution, they strug-

gle with sparse or fragmented data and fail under out-of-distribution interventions (Pearl, 2009; Peters et al., 2017).

**Foundation Models as World Models.** Recent work explores harnessing large foundation models, including LLMs, to simulate environments for decision-making (Gao et al., 2024; Hao et al., 2023; Liu et al., 2024; Yang et al., 2024; Xie et al., 2024; Wang et al., 2024b; Zhou et al., 2024; Cherian et al., 2024). While even partially correct models can boost sample efficiency in MBRL, they frequently produce biased or inconsistent trajectories when asked to simulate real-world systems (Vafa et al., 2024). Subtle inaccuracies and noise compounds over time (Lambert et al., 2022), and their limited capacity to systematically track multi-faceted interactions undermines their reliability for complex, real-world simulations.

**LLM-Coded Simulations.** Several methods use LLMs to *generate* environment code. OMNI-EPIC and GenSim create open-ended environments for agent learning and do not aim to mirror real systems (Faldor et al., 2024; Wang et al., 2024a). WorldCoder (Tang et al., 2024) is a notable work aimed at MBRL for deterministic, discrete logic but is only partially calibrated to real-world evidence through refinement. Consequently, it lacks robust mechanisms for handling stochastic processes, partial observations, or principled numerical parameter inference.

**Hybrid Digital Twins.** Hybrid digital twins combine mechanistic models with data-driven corrections to capture unmodeled dynamics (Holt et al., 2024b), but they often assume continuous physical processes and do not fully generalize to discrete, stochastic, or heavily modular domains.

**Comparison with Prior Work.** In Table 1, we summarize how G-Sim aligns with and diverges from these approaches. G-Sim's novelty lies in merging domain-knowledge-based structural priors with flexible, gradient-free calibration of discrete or stochastic modules. This combination accommodates fragmented data and enables robust out-of-distribution stress-testing and *policy* interventions, bridging the gap between purely data-driven and purely LLM-generated simulators.

*Table 1.* Comparison of G-Sim with representative methods. ✗ indicates a missing feature, ✓ a supported one, while lim denotes partial fulfillment. *Structural Prior* refers to uncovering simulator topology from domain knowledge. *Data-form Flexible* indicates support for continuous, discrete, or stochastic processes. *Emp. Calib.* stands for data-driven parameter calibration. *OOD Stress* means robust performance under out-of-distribution scenarios.

| Method | Structural Prior | D-form-Flex. | Emp. Calib. | OOD Stress |
|---|---|---|---|---|
| *Hybrid Digital Twins* | ✓ | ✗ | ✗ | ✓ |
| *WorldCoder* | ✓ | ✗ | lim | ✓ |
| *Data-Driven World Model* | ✗ | lim | ✓ | ✗ |
| **G-Sim (Ours)** | ✓ | ✓ | ✓ | ✓ |

# 5. Experiments and Evaluation

In this section, we evaluate G-Sim to verify that it can generate simulators with higher fidelity than existing discovery or data-driven world models. Our experiments use both GFO and SBI for calibration.

**Benchmark Environments.** We evaluate G-Sim on three real-world-inspired simulation tasks that together capture (1) stochastic transitions, (2) rich, discrete state updates, and (3) partially observed states. Each task provides a dataset of state-action trajectories and a textual description of the environment, sampled from a carefully hand-designed simulator. First, our **COVID-19** epidemiological simulation extends classical compartmental frameworks (Cooper et al., 2020; AlQadi & Bani-Yaghoub, 2022) to incorporate discrete, stochastic transitions; it tracks populations moving across compartments (e.g., susceptible, infectious, recovered). Second, the **Supply Chain** environment is based on the well-known "beer game" (Sterman, 1989), which simulates demand fluctuations and the resulting bullwhip effect across multiple stages (retailer, wholesaler, distributor, manufacturer). This environment is partially observed because orders are processed in a pipeline, causing delays and uncertainty around incoming shipments. Finally, the **Hospital Bed Scheduling** environment simulates patient arrivals for three different diseases into a hospital with a finite number of ICU and standard care beds (Green, 2006; Koizumi et al., 2005; Brailsford, 2007). Each disease has its own arrival rate, length-of-stay distribution, and daily mortality probability, leading to partial observability and discrete, stochastic transitions (e.g., admissions, discharges, and deaths). We provide a detailed discussion of these tasks and their datasets in Appendix C.

**Evaluation Metrics.** We adopt the Wasserstein distance as our primary evaluation metric. From each initial state in the held-out test set, we simulate $N$ trajectories under both the *ground-truth* and *comparison* simulators, then compute the Wasserstein distance between these two sets of next-state samples. We repeat this for all initial states in the test set and average the distances, thereby measuring how well each simulator reproduces the ground-truth distribution. We run five independent trials with different seeds, reporting the mean and 95% confidence intervals. Further details are provided in Appendix H.

**Benchmark Methods.** We compare G-Sim against a diverse set of approaches covering three main categories. First, *data-driven world models* learn environment dynamics from state-action trajectories without explicit structural priors: we employ a recurrent neural network (**RNN**) (Rumelhart et al., 1986), a competitive causal **Transformer** (Melnychuk et al., 2022), and a neural ordinary differential equation with action inputs (**DyNODE**) (Chen et al., 2018; Alvarez et al., 2020). Second, *equation discovery* methods aim to uncover

*Table 2.* Test Wasserstein distance (lower is better) on three environment-generation tasks, averaged over five random seeds ($\pm$ denotes 95% CIs). Light-blue shading highlights our method.

| Method | Test Wasserstein distance ($\downarrow$) | | |
| --- | --- | --- | --- |
| | COVID-19 | Supply Chain | Hospital Beds |
| DyNODE | $65.1 \pm 2.21$ | $38.3 \pm 0.40$ | $231 \pm 0.14$ |
| SINDy | $23.9 \pm 0.40$ | $18.2 \pm 0.24$ | $199 \pm 0.04$ |
| RNN | $16.7 \pm 1.61$ | $9.71 \pm 2.21$ | $199 \pm 2.49$ |
| Transformer | $3.30 \pm 0.15$ | $2.29 \pm 0.06$ | $199 \pm 0.25$ |
| Genetic Program | $63.6 \pm 7.64$ | $30.7 \pm 1.41$ | $231 \pm 0.04$ |
| G-Sim-ES Abl. ZeroShot | $1.17 \pm 0.71$ | $2.63 \pm 2.79$ | $102 \pm 1.01$ |
| G-Sim-ES Abl. ZeroShotOptim | $0.469 \pm 0.107$ | $9.89 \pm 15.3$ | $103 \pm 2.06$ |
| **G-Sim – SBI** | $\mathbf{0.351 \pm 0.094}$ | $\mathbf{1.22 \pm 1.68}$ | $\mathbf{5.24 \pm 2.70}$ |
| **G-Sim – ES** | $0.405 \pm 0.060$ | $\mathbf{1.55 \pm 1.39}$ | $101 \pm 17.4$ |

mechanistic or symbolic equations directly from data: we use **SINDy** (Brunton et al., 2016) and a **Genetic Program** (De Rainville et al., 2012) that searches for symbolic expressions via evolutionary algorithms. Moreover we compare two variants of G-Sim, of G-Sim with GFO of Evolutionary Strategies (ES) (**G-Sim – ES**) and G-Sim with simulation based inference (SBI) (**G-Sim – SBI**). Lastly, to isolate the contributions of G-Sim's iterative refinement, we include two ablations: (**G-Sim-ES Abl. ZeroShot**) uses the LLM to generate simulator code *once* (with no parameter calibration), and (**G-Sim-ES Abl. ZeroShotOptim**) applies gradient-free optimization only to numerical parameters (without adjusting the structural design). All baselines share the same training and evaluation splits, and detailed implementation and hyperparameter settings are given in Appendix D.

# 6. Main Results

We evaluated all benchmark methods across the three environments, with results tabulated in Table 2. G-Sim consistently achieves the lowest Wasserstein distance on the held-out test data, indicating that its generated simulators model the ground-truth system dynamics with the highest fidelity. The performance gap is particularly pronounced in the complex Hospital Bed Scheduling task, where data-driven methods struggle significantly.

Beyond predictive accuracy, we demonstrate G-Sim's unique capability to answer "*what if?*" questions involving *policy* or structural interventions that lie outside the training data distribution. These insight experiments showcase G-Sim's ability to generalize to novel scenarios and inform decision-making in complex systems, a task for which other methods lack a direct mechanism.

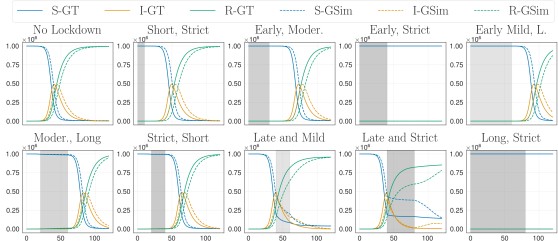

*Figure 2.* **Lockdown intervention on COVID-19 SIR.** We impose a temporary lockdown (grey rectangles with opacity proportional to intensity) by scaling the infection rate $\beta \mapsto \alpha\,\beta$ for different start/end times and $\alpha \in \{0.05, 0.1, 0.15, 0.3\}$. The *solid lines* are ground truth; *dashed lines* are G-Sim's predictions. G-Sim correctly adapts to these unseen interventions, maintaining predictive performance.

### 6.1. Insight Experiments and *Policy* Interventions

**SIR Lockdown Interventions.** We first examine a COVID-19 scenario where a lockdown multiplier $\alpha \in [0, 1]$ scales the infection rate $\beta$ for a specified interval $[t_{\text{start}}, t_{\text{end}}]$. This models the effect of a temporary lockdown by reducing the rate of new infections. We impose multiple lockdown scenarios by varying $\alpha$ and the lockdown duration to assess whether G-Sim can accurately replicate the resulting infection trajectories.

Figure 2 compares the ground truth with G-Sim's predictions under different lockdown intensities ($\alpha = 0.05, 0.1, 0.15, 0.3$). Despite not encountering lockdown events during training, G-Sim successfully captures the delayed and reduced infection peaks corresponding to the imposed interventions. In contrast, the other baselines fail to incorporate this structural change, rendering them inapplicable for such an analysis.

***Policy* Optimization.** Next, we demonstrate G-Sim's utility for *policy* optimization by searching over a discrete set of interventions in the Hospital Bed Scheduling task. The interventions combine *capacity expansion* ($\Delta B$, additional beds) and *lockdown scheduling* ($\tau$, the start day of a fixed 20-day lockdown). The cost function to minimize is:

$$\text{Cost} = \text{Overflow} + 10 \times \Delta B + 20 \times \text{lockdown\_duration},$$

where Overflow is the number of patients exceeding bed capacity.

We optimize over the grid $(\tau, \Delta B) \in \{0, 5, \ldots, 95\} \times \{0, 500, \ldots, 9500\}$. Table 3 shows that the best policy found using the G-Sim simulator is nearly identical to the true optimal policy, demonstrating that policies optimized with G-Sim are effective and transferable to the real environment.

**Supply Chain: Resource Optimization.** We explore resource optimization in the supply chain environment by

*Table 3.* Comparison of optimal policies for Ground Truth and G-Sim environments. We report the lockdown start day $\tau$, additional beds $\Delta B$, and total cost. G-Sim identifies a policy that closely approximates the ground truth's optimal strategy with minimal cost deviation.

| Method | $\tau^*$ | $\Delta B^*$ | Cost |
|---|---|---|---|
| Ground Truth Best | 10 | 2500 | 29,274 |
| G-Sim Best | 15 | 2500 | 32,703 |

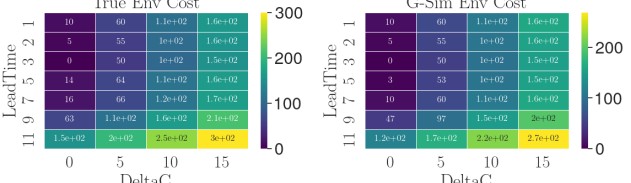

*Figure 3.* **Supply-chain resource optimization.** Heatmaps of total cost $\text{Cost}(\Delta C, \ell)$ as a function of extra capacity $\Delta C$ and lead time $\ell$. *Left*: Ground truth. *Right*: G-Sim. Both heatmaps exhibit similar cost landscapes, demonstrating that G-Sim effectively models the cost trade-offs.

varying both $\Delta C$ (extra warehouse capacity) and $\ell$ (lead time). Figure 3 presents heatmaps of the total cost as a function of $\Delta C$ and $\ell$ for both the ground truth and G-Sim environments. The striking similarity in the global structure of the cost landscapes indicates that G-Sim accurately captures the trade-offs between capacity expansion and shipping delays. Consequently, the optimal regions in the G-Sim environment align closely with those in the ground truth, affirming G-Sim's reliability for strategic resource allocation.

**Supply Chain: Varying Lead Times.** In Appendix I we further stress-test the supply chain simulator by introducing varying lead times $\ell$, which were not explicitly present during training, to test the simulator's ability to handle unseen delays.

## 7. Discussion and Conclusion

We introduced **G-Sim**, a novel framework that automates simulator construction by synergizing LLM-driven structural design with rigorous empirical calibration. G-Sim's key innovation is its flexibility, offering a choice of powerful, **likelihood-free** and **gradient-free** techniques: **Gradient-Free Optimization** for direct parameter estimation or **Simulation-Based Inference** for Bayesian posterior inference. This hybrid approach overcomes the critical limitations of purely data-driven models (poor OOD generalization) and purely LLM-generated ones (lacking empirical grounding).

G-Sim's iterative refinement loop co-evolves a simulator's

structure and parameters, driven by diagnostic feedback, to achieve both causal plausibility and empirical alignment. The option to use SBI is particularly powerful for applications requiring principled uncertainty quantification. However, a key technical nuance must be appreciated: SBI's guarantees assume a *correct model structure*. Within G-Sim's search process, SBI posteriors correctly reflect parameter uncertainty *given* a proposed structure, but do not capture the overarching structural uncertainty. Modeling this structural uncertainty explicitly is a vital frontier for future research.

The practical implications of G-Sim are significant. By producing intervenable simulators with plausible generalization, G-Sim facilitates robust "what if?" analyses in critical domains like epidemic planning, supply chain management, and healthcare logistics. Our experiments demonstrate that G-Sim not only replicates observed dynamics but also accurately predicts system behavior under novel conditions, offering a powerful tool for *policy* evaluation and design.

Limitations remain, notably the scalability to extremely high-dimensional systems and ensuring the LLM's proposed structures are sufficiently diverse. We discuss these, along with ethical considerations and future work, in Appendix G.

In conclusion, G-Sim offers a flexible and robust path towards building more accurate, causally consistent, and uncertainty-aware simulations. By integrating structural reasoning with data-driven calibration, it marks a significant step forward in automatic simulation generation, enabling deeper insights and better decisions in complex systems.

## Acknowledgements

We extend our gratitude to the anonymous reviewers, area and program chairs, and members of the van der Schaar lab for their valuable feedback and suggestions. We also thank Daniel Gedon for their insightful comments and suggestions that ultimately improved this work. SH & ML gratefully acknowledge the sponsorship and support of AstraZeneca. AB acknowledges funding from Eedi. This work was supported by Azure sponsorship credits granted by Microsoft's AI for Good Research Lab and by Microsoft's Accelerate Foundation Models Academic Research Initiative.

## Impact Statement

Our approach (G-Sim) can enhance decision-making in fields like healthcare, logistics, and climate science by automating simulator construction from sparse data and domain knowledge, enabling safer, cost-effective "what if?" analysis. However, reliance on LLM-generated structures or calibration with biased data could risk misleading outcomes if not properly validated. Appropriate oversight, domain expertise, and transparency regarding assumptions are crucial for responsible use, ensuring these simulators support ethical and beneficial real-world applications.

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

# Appendix

## Table of Contents

# A. Extended Related Work

Automatically building general-purpose simulators has been a longstanding challenge in machine learning and simulation research. Below, we position our proposed approach (*G-Sim*) within several primary research streams—including data-driven world models, foundation-model-based and LLM-coded world models, and hybrid digital twins—highlighting how G-Sim uniquely addresses their respective limitations for real-world simulation-building. We then expand on additional threads such as equation discovery, procedural content generation, and active learning.

## A.1. Data-Driven World Models

A substantial body of work in model-based reinforcement learning (MBRL) focuses on learning parametric approximations of environment dynamics purely from data (Ha & Schmidhuber, 2018; Hafner et al., 2019; Alonso et al., 2023; Micheli et al., 2023; Hafner et al., 2023; Ding et al., 2024; Bruce et al., 2024). Typically referred to as *world models*, these methods leverage neural networks to predict transitions and rewards, often in a compressed latent space, to improve sample efficiency and planning performance. However, as discussed in our main paper, such purely data-driven approaches are ill-suited to broader system-level questions such as structural interventions or scenario analyses (Pearl, 2009; Peters et al., 2017; Kacprzyk et al., 2024). They also rely heavily on large and representative datasets, which are often unavailable in real-world domains that suffer from sparse or fragmented data, and they tend to break down under out-of-distribution conditions.

## A.2. Foundation Models as World Models

Recent work explores using large foundation models (including LLMs) as world models for decision-making (Gao et al., 2024; Hao et al., 2023; Liu et al., 2024; Yang et al., 2024; Xie et al., 2024; Wang et al., 2024b; Zhou et al., 2024; Cherian et al., 2024). These models can plausibly "role-play" entire environments given enough textual context, sometimes boosting sample efficiency in MBRL tasks. However, they frequently produce biased or inconsistent trajectories when asked to simulate real-world systems (Vafa et al., 2024). The limited capacity of language models to systematically track multi-faceted, time-evolving interactions leads to compounding errors (Lambert et al., 2022), undermining reliability in complex settings.

## A.3. LLM-Coded Simulations

Another line of research employs LLMs to *generate environment code* for simulation. For example, *OMNI-EPIC* and *GenSim* build open-ended Pybullet (Coumans & Bai, 2016–2021) environments by synthesizing code in physics engines (Faldor et al., 2024; Wang et al., 2024a). While effective for agent exploration, these methods rarely aim to *mirror real-world systems* or align with empirical data. In contrast, the recent *WorldCoder* approach (Tang et al., 2024) does generate environment code from textual descriptions, supporting some iterative refinement with real-world evidence. However, it remains limited to deterministic logic, lacks robust mechanisms for handling partial observability or stochasticity, and cannot systematically infer quantitative parameters from real data.

## A.4. Hybrid Digital Twins and Mechanistic Models

Another relevant trend involves *hybrid digital twins*, which combine known mechanistic or physical processes with data-driven correctors (Holt et al., 2024b;c). These approaches excel in domains where a partial physical law or differential equation is known and can be complemented by a learned residual neural module. Yet they generally require substantial domain expertise to specify the underlying differential equation or other rigid differentiable forms, and do not generalize well to systems with discrete, stochastic events or partially specified modules. Likewise, purely mechanistic *equation-discovery* approaches (e.g., SINDy (Brunton et al., 2016) or PDE-based symbolic regression) can uncover closed-form ODEs from data, but they often fail when the system is highly modular, partially observed, or contains discrete jumps.

## A.5. Procedural Content Generation & Environment Simulation

Procedural generation techniques have long been used in game design to create diverse levels or scenarios automatically (Shaker et al., 2016; Khalifa et al., 2020). LLM-based text-to-environment pipelines (Li et al., 2024) can also create interesting scenarios from narrative descriptions. However, these approaches rarely incorporate real-world evidence or aim to produce *data-calibrated* transition dynamics. By contrast, G-Sim seeks not only to produce environment code but also to *calibrate* parameters against real (potentially fragmented) data, resulting in an environment that is both plausible and

empirically grounded.

### A.6. Active Learning and Iterative Refinement

Active learning and simulation-based inference methods (Cranmer et al., 2020; Settles, 2009; Pouplin et al., 2024) have been utilized to reduce data requirements and improve simulator fidelity. Gradient-free optimizers such as evolutionary strategies (Sehnke et al., 2010; Toklu et al., 2023; Holt et al., 2024a) can handle non-differentiable or stochastic objectives by repeatedly simulating candidate parameter sets. While effective for calibration, these methods alone assume a fixed simulator structure; they do not address *how to build or refine* the structural design itself. G-Sim integrates LLM-based structural generation *and* flexible calibration in an iterative loop, allowing the simulator's topology to be revised whenever diagnostics indicate insufficient fidelity or domain compliance.

### A.7. Comparison with Prior Work

Table 1 summarizes key features of G-Sim in comparison to representative approaches. Purely data-driven (or purely LLM-driven) pipelines struggle with complex, partially observed, or discrete systems; they also lack a mechanism for systematically merging domain knowledge with real data. Hybrid digital twins require continuous or well-defined mechanistic equations, while purely code-generating approaches often do not incorporate rigorous calibration steps. In contrast, *G-Sim* addresses these gaps by:

1. Leveraging **LLM-guided structural proposals** to capture domain-appropriate topologies and submodules,

2. Using a choice of **flexible, likelihood-free calibration methods** (GFO or SBI) to tune parameters against real data,

3. Employing an **iterative refinement loop** that diagnoses discrepancies and re-queries the LLM for adjustments.

This hybrid, compositional approach unlocks new capabilities for real-world simulator creation, where partial observations, non-differentiable transitions, and out-of-distribution *policy* interventions are present.

**Key Differences and Contributions.**    In sum, no prior framework integrates LLM-based structural reasoning, flexible parameter inference, and an iterative refinement mechanism to produce high-fidelity, data-grounded simulators in a *single* pipeline. By unifying domain knowledge and empirical calibration, G-Sim offers robust scenario planning, *policy* intervention testing, and OOD stress-testing in ways that purely data-driven or purely LLM-generated simulators cannot. We believe this synthesis of large language models, GFO, and SBI opens a promising direction toward truly *automatic environment generation* for high-stakes real-world domains.

## B. Additional Theoretical Considerations and Implementation Details

This appendix provides a deeper look at the theoretical assumptions, identifiability properties, and practical implementations that underpin our *G-Sim* framework. Section B.1 discusses the nonzero coverage assumption for the LLM's structural proposals, including potential failure modes. Section B.2 expands on structural identifiability and equivalence classes. Section B.2 provides detailed guidelines for constructing the diagnostics function $\mathrm{Diag}$. Finally, Section B.3 illustrates how to ensure sufficiently rich prompts to maintain broad coverage from the LLM.

### B.1. LLM Coverage Assumption

In the main text, we assume:

$$p_{\mathrm{LLM}}(\lambda \mid \mathcal{K}) \ \geq \ \alpha \ > \ 0 \quad \text{for all } \lambda \in \mathcal{C}^*(\mathcal{S}),$$

where $\mathcal{C}^*(\mathcal{S}) \subseteq \mathcal{C}(\mathcal{S})$ is a set of feasible structural configurations that adequately represent the real-world system (up to a certain approximation). This requirement guarantees that an iterative procedure—repeatedly prompting the LLM for revised structures—can eventually propose a structure that is "close enough" to the true environment to support accurate inference.

**When Coverage Might Fail.** Several practical factors can undermine this assumption:

- **Under-trained or Domain-Mismatched LLM.** If both the LLM's training corpus and the explicit knowledge $\mathcal{K}$ lack relevant domain knowledge or code examples (e.g., queueing systems, compartmental epidemic models), then many relevant configurations may be assigned near-zero probability.

- **Excessive Prompt Constraints.** Overly restrictive or poorly designed prompts can steer the LLM away from generating diverse substructures.

- **Novel or Rarely Documented Submodules.** If the true system relies on custom or highly specific domain processes that rarely appear in public text corpora, the LLM might not "know" how to compose those substructures.

In such cases, either prompt-engineering $\mathcal{K}$ or partial fine-tuning can help restore coverage (see Appendix B.3 and Appendix E.4). Ultimately, if no structural representation in $\mathrm{supp}(p_{\mathrm{LLM}}(\lambda \mid \mathcal{K}))$ is even *approximately* valid, no LLM approach can succeed.

**Approximate vs. Exact Structural Matches.** In real applications, there may be no perfectly correct simulator structure. Instead, $\mathcal{C}^*(\mathcal{S})$ can be a collection of *good enough* or *functionally sufficient* structures (e.g., ignoring minor confounders, or grouping certain processes together). If one such structure is proposed by the LLM, subsequent calibration of $\omega$ should suffice for robust predictions.

### B.2. Structural Identifiability

**Definition.** *Structural identifiability* typically refers to whether the mapping

$$(\lambda, \, \omega) \ \mapsto \ \Big\{ \text{possible data distributions from the simulator} \Big\}$$

is injective (one-to-one) in some relevant sense. In simpler terms: if two distinct structures or parameterizations generate *identical* distributions over observable data, they are *structurally unidentifiable* from an empirical standpoint.

**Why This Matters.** In our context, an unidentifiable simulator might mean that even *perfect* (infinite) data is insufficient to discriminate between different submodule configurations. One example is an environment that can be explained equally well by either an $M/M/c$ (Kermack et al., 1997) queue with a certain arrival rate or a more complex arrival process with time-varying rates, as long as the overall distribution of arrival times matches. Under such conditions, the refinement loop may not converge to a unique $\lambda$, and we might have to choose a heuristic such as Occam's razor (Sammut & Webb, 2011).

**Partial Identifiability and Equivalence Classes.** In practice, especially with partially observed or unpaired data, we often only identify *equivalence classes* of structures. That is, the set of candidate $\lambda$ can remain multi-modal, with multiple plausible subgraphs explaining the data. In such cases, the final output might be a *distribution* or *set* over competing submodule graphs. One can still proceed to do *policy* evaluations or forward simulations by sampling from that distribution over possible structures.

**Remedies.** Two approaches can mitigate structural unidentifiability:

1. **Additional Domain Knowledge.** Imposing explicit structural constraints (e.g., "patient severity depends on risk factors, not on staff scheduling") can prune the space of submodules and break identifiability symmetries.

2. **Refined Diagnostics.** Using more fine-grained checks (e.g., dissecting arrival patterns by time-of-day or day-of-week) can help discriminate between seemingly identical structures.

Ultimately, structural identifiability in highly complex systems remains an open challenge. We thus recommend domain experts remain in the loop to verify or reject certain submodule proposals as needed.

### Design of the Diagnostics Function

The refinement loop (Section 3.3) in the main text relies on a diagnostic function

$$\mathrm{Diag}\big(\lambda, \omega^*\big) \;\mapsto\; \mathbb{R}_{\geq 0}$$

to measure goodness-of-fit and highlight structural gaps. This function, which we denote as $d$, aggregates various signals indicating how well the current simulator $(\lambda, \omega^*)$ matches the real system. Below, we offer more details on how to implement $\mathrm{Diag}$ in a way that is both flexible and tractable, followed by how it is realised in practice.

PREDICTIVE CHECKS (PPC)

A standard tactic in statistics is to compare *simulated* data against real data. Specifically:

1. **Generate Simulated Data:** For the current structure $\lambda$ and calibrated parameters $\omega^*$, forward-simulate trajectories $x^{(m)} \sim f\big(\cdot\,; \lambda, \omega^*\big)$. If using SBI, one would draw posterior samples $\omega^{(m)} \sim \widehat{p}(\omega \mid \mathcal{D})$ and simulate for each.

2. **Compute Summary Statistics:** Calculate relevant summary statistics $T\big(x^{(m)}\big)$ (e.g., mean arrival rates, bed occupancy distributions, trajectory-wise errors).

3. **Measure Discrepancy:** Compare the distribution of $T\big(x^{(m)}\big)$ to $T\big(\mathcal{D}_{\mathrm{real}}\big)$ using suitable distance metrics.

This predictive discrepancy, $\delta_{\mathrm{predictive}}$, can be defined using metrics like:

- **Wasserstein distance ($W_1$)** (Kantorovich, 1960): Measures the 'work' required to transform one distribution into another.

- **Maximum Mean Discrepancy (MMD)** (Gretton et al., 2012): A kernel-based distance often used in generative modeling.

- **Mean Squared Error (MSE)**: Useful for specific component outputs.

- **Kullback-Leibler (KL) divergence** (Shlens, 2014): Measures information loss.

One might define $\delta_{\mathrm{predictive}} = \mathrm{WASS}\Big(\big\{T(x^{(m)})\big\},\, T(\mathcal{D}_{\mathrm{real}})\Big)$, or a weighted sum over multiple statistics.

PARAMETER UNCERTAINTY CRITERIA (FOR SBI)

When using Simulation-Based Inference (SBI) (Cranmer et al., 2020), the posterior distribution $\widehat{p}(\omega \mid \mathcal{D})$ itself can signal model misspecification. A very large variance or multiple distinct modes might indicate that the current structure $\lambda$ fails to "pin down" key parameters. One could define a diagnostic like:

$$\mathrm{Diag}_{\mathrm{Var}} \;=\; \sum_{i=1}^{d} \mathbf{1}\Big(\mathrm{Var}\big[\omega_i\big] > \tau_i\Big)$$

where $\tau_i$ is a threshold for the variance of each parameter $\omega_i$.

STRESS TESTING & PLAUSIBILITY CHECKS

To assess generalization and robustness, we can check the simulator's behaviour under out-of-distribution (OOD) scenarios or against known domain constraints (e.g., unit tests (Li & Yuan, 2024)).

$$\delta_{\text{domain}} = \sum_{j=1}^{J} \mathbf{1}\Big(\text{Simulator yields implausible outputs under scenario } \Omega_j\Big) + \sum_{k=1}^{K} \mathbf{1}(\text{Rule } k \text{ violated})$$

Here, each $\Omega_j$ is a hypothetical stress test (e.g., a sudden surge in patient arrivals), and "implausible outputs" might include negative states or constraint violations.

COMBINING DIAGNOSTICS AND SETTING THE THRESHOLD ($\epsilon$)

The individual diagnostic components ($\delta_{\text{predictive}}$, $\delta_{\text{domain}}$, etc.) can be aggregated into a single scalar score, $d = \text{Diag}(\lambda, \omega^*)$. A common approach is a weighted sum:

$$\text{Diag}(\lambda, \omega^*) \;=\; w_1\, \delta_{\text{predictive}} \;+\; w_2\, \delta_{\text{domain}} \;+\; \cdots$$

with weights $w_i \geq 0$. This score $d$ is then compared against a predefined threshold $\varepsilon$. If $d > \varepsilon$, refinement is triggered. When refinement occurs, the specific failing components inform the generation of textual feedback, guiding the LLM's next proposal.

PRACTICAL IMPLEMENTATION IN G-SIM

In our implementation, the diagnostic function ($d$) primarily relies on **Posterior Predictive Checks**. We calculate both the **Wasserstein distance** and **Mean Squared Error (MSE)** between simulated trajectories (generated using parameters found via calibration) and a held-out validation dataset. These metrics provide a quantitative measure of how well the simulator's outputs match empirical observations.

The **Wasserstein distance** is used as the primary metric ($d = \delta_{\text{predictive}} = W_1$) for ranking different simulator structures generated by the LLM within each iteration and for tracking overall progress. However, the feedback provided to the LLM for refinement is more detailed. It includes not only the overall Wasserstein score but also the **MSE broken down per dimension** of the simulator's state space. This allows the LLM to understand *which specific parts* of the simulator are performing poorly (e.g., underestimating 'infected' counts or overestimating 'inventory' levels) and propose more targeted structural changes.

Regarding the **threshold ($\epsilon$)**, our framework uses an **implicit approach**. We do not set a specific numerical value for $\epsilon$. Instead, the refinement loop terminates based on two conditions:

1. **Iteration Limit:** A maximum number of G-Sim iterations (or 'generations') is defined in the configuration. This sets a hard limit on the computational budget.

2. **Early Stopping:** We monitor the best Wasserstein distance achieved in each generation. If this score does not improve for a predefined number of consecutive generations (a 'patience' parameter), the loop terminates early.

This strategy means the refinement process continues as long as it is finding significantly better simulator structures within its budget, effectively setting $\epsilon$ based on observed progress rather than a fixed *a priori* value. When the loop terminates, the simulator with the lowest achieved Wasserstein distance is selected as the final output.

## B.3. Prompt-Engineering for Broader LLM Coverage

Since the LLM proposals $\lambda$ govern the success of the refinement loop, *prompt-engineering* (Sahoo et al., 2024) is crucial for ensuring that:

1. The LLM has *enough freedom* to propose submodules beyond an initial guess.

2. The domain knowledge $\mathcal{K}$ is well-articulated so that the LLM can access relevant structural ideas.

3. Textual feedback from the diagnostic function $\text{Diag}$ is expressed in a sufficiently precise and instructive manner, so that the LLM can respond by refining the submodule that is problematic with reasonable probability.

**Concrete Prompting Steps.** In Appendix E.4 we provide the prompt templates that we use in our experiments.

In case one wished to explore multiple structural variants, they could sample multiple completions from the LLM (by adjusting temperature or top-$k$ sampling), score each proposed structure via Diag and retain the best structure(s) for iteration. This approach follows an *evolutionary search* spirit in the space of submodule graphs. In practice, controlling sampling temperature or employing specialized generation routines (e.g., "chain-of-thought" (Wei et al., 2022) prompts that systematically reason about possible modules) can yield better coverage than a single pass (Wei et al., 2022). In any case, we leave these iterations of our framework to future work.

**Limitations.** Even with carefully engineered prompts, the LLM might produce repetitive or irrelevant suggestions. If the system remains stuck in a poor local optimum (see Appendix B.1), one might want to invoke direct expert intervention, manually expanding $\mathcal{K}$, providing a new submodule explicitly or specifying more explicit constraints. Indeed, while the *iterative refinement loop* can help, it is not guaranteed to find a global optimum if the LLM's proposal distribution or domain constraints are too limited.

### B.4. Limitations of Simulation-Based Inference in G-SIM

While Simulation-Based Inference (SBI) provides a principled route to parameter estimation and uncertainty quantification (Cranmer et al., 2020), its naive adoption inside G-SIM requires special care because several of SBI's core theoretical assumptions are violated once the model structure is itself unknown.

**The Model Mismatch Problem.** Classical guarantees—e.g. posterior consistency, well-calibrated credible sets and successful simulation-based calibration (SBC)—all rely on the simulator $f(\cdot; \lambda^*, \omega)$ being correctly specified and fixed (Cranmer et al., 2020; Talts et al., 2020). In G-SIM we deliberately explore candidate structures $\lambda^{(g)}$ generated by an LLM; whenever $\lambda^{(g)} \neq \lambda^*$ we enter a misspecified regime. Theory shows that Bayesian posteriors can become biased or over-confident under misspecification (Nott et al., 2024), and recent empirical studies demonstrate the same failure modes for neural SBI algorithms (Hermans et al., 2022; Kelly et al., 2025). Hence the learned posterior $q_\phi(\omega \mid y, \lambda^{(g)})$ may still converge—but not to the distribution that reflects our true uncertainty under the correct model.

**Observational Signatures and Future Directions.** Empirically, severe misspecification often manifests as excessively broad or oddly-shaped posteriors (Hermans et al., 2022). Posterior-shape diagnostics based on entropy or variance can flag such cases in amortised SBI (Schmitt et al., 2024; Wehenkel et al., 2024). Integrating these metrics into the Diag function would give G-SIM an automatic "early-warning" system for structural defects.

Looking ahead, three research directions appear particularly promising:

- **Bayesian Model Averaging.** Maintaining a weighted ensemble of plausible structures and marginalising over them tackles structural as well as parametric uncertainty (Spurio Mancini et al., 2023; Wehenkel et al., 2024).

- **Robust Guarantees under Approximate Correctness.** Extending the analysis of Nott et al. (2024) to simulators, or developing divergence-based bounds, could yield milder but still useful uncertainty guarantees even when $\lambda^{(g)}$ is only approximately correct (Kelly et al., 2025).

- **Active Learning for Structure Discovery.** "Query-by-disagreement" strategies that maximise posterior divergence between rival structures can accelerate the search process (Griesemer et al., 2024).

Equipping G-SIM with these tools would move it beyond the single-model assumption inherited from classical SBI and towards a principled treatment of structural uncertainty.

# C. Benchmark Dataset Environment Details

We evaluate our proposed approach on a suite of real-world-inspired benchmark tasks, each designed to reflect critical properties of real system dynamics such as stochastic updates, discrete state transitions, and partial observability. These benchmark environments serve as testbeds for assessing how well different simulator-building frameworks capture complex behaviors, align with empirical data, and generalize to interventions that are absent from the training distribution. Here, we describe in detail the structure, parameters, and data generation procedures for each environment, enabling reproducibility and clarifying the unique modeling challenges each setting presents.

## C.1. COVID-19 SIR Environment

Our **COVID-19 SIR** environment is inspired by the classic compartmental SIR framework and extends modern discrete-time variants (Cooper et al., 2020; AlQadi & Bani-Yaghoub, 2022) by modeling discrete, stochastic transitions of individuals among three key health states:

- **S (Susceptible)**: Number of individuals who are healthy but can become infected.

- **I (Infectious)**: Number of currently infected individuals capable of transmitting the pathogen.

- **R (Recovered)**: Number of individuals who have recovered (or are otherwise removed) from the disease, and cannot become infected again.

**Simulation code and parameters.** Listing 1 illustrates the core update procedure. The simulator maintains two parameters, $\beta$ and $\gamma$, which govern transition rates:

- $\beta$ (*base transmission rate*): Higher $\beta$ implies that infections spread more aggressively in the population.

- $\gamma$ (*daily recovery probability*): Individuals in the `I` compartment recover and transition to `R` at a Binomial rate characterized by $\gamma$.

At each step $t$, the environment receives the current state $s_t = \{S_t, I_t, R_t\}$ and an *action* $a_t$. While classical epidemic modeling may treat non-pharmaceutical interventions (NPIs) or *policy* actions as external controls, this particular version does not yet incorporate any direct effect of `action`. However, the simulator is extensible so that future versions can incorporate lockdowns, vaccination campaigns, or other interventions by modifying the effective $\beta$ or $\gamma$ in a time-varying manner.

*Listing 1.* Core step function of the COVID-19 SIR simulator. The environment updates $(S, I, R)$ by sampling the number of new infections and new recoveries from a Binomial distribution.

```
class SimulatorStep:
    def __init__(self):
        # Default parameters: beta=0.5, gamma=0.1
        self.parameters = np.array([0.5, 0.1], dtype=float)

    def step(self, state: dict, action: int) -> dict:
        """
        Performs a single-step update for a discrete-time SIR model.
        """
        # Extract compartments
        S = state["S"]
        I = state["I"]
        R = state["R"]

        # Unpack parameters
        beta, gamma = self.parameters

        # Compute total population
        N = S + I + R
        if N <= 0:
            return {"S": 0, "I": 0, "R": 0}  # Degenerate case

        # Deterministic infection probability (rate) from susceptible to infected
        prob_infection = 1.0 - np.exp(-beta * I / N)
        prob_infection = np.clip(prob_infection, 0.0, 1.0)

        # New infections ~ Binomial(S, prob_infection)
        new_infections = safe_binomial(S, prob_infection)
        # New recoveries ~ Binomial(I, gamma)
        new_recoveries = safe_binomial(I, gamma)

        # Update compartments
        next_S = S - new_infections
        next_I = I + new_infections - new_recoveries
        next_R = R + new_recoveries
```

```
    return {"S": next_S, "I": next_I, "R": next_R}
  ...
```

**Discrete-time updates.** Unlike a continuous-time SIR model that uses ordinary differential equations, we adopt a discrete-time approach. At each discrete time step:

$$\text{new\_infections} \quad \sim \text{Binomial}\Big(S_t,\ 1 - e^{-\beta \cdot \frac{I_t}{N_t}}\Big),$$

$$\text{new\_recoveries} \quad \sim \text{Binomial}\big(I_t,\ \gamma\big),$$

$$S_{t+1} = S_t - \text{new\_infections}, \quad I_{t+1} = I_t + \text{new\_infections} - \text{new\_recoveries}, \quad R_{t+1} = R_t + \text{new\_recoveries}.$$

Here, $N_t = S_t + I_t + R_t$ is the total population at time $t$. When the population is nonzero, the probability of infection among susceptible individuals is modeled by $1 - e^{-\beta(I_t/N_t)}$, reflecting a common deterministic approximation to the force of infection.

**Stochastic transitions.** The model draws random samples for new infections and recoveries using a `Binomial` function; hence the counts of new infections or recoveries vary across simulations, even with the same initial condition and parameters. This yields a more realistic depiction of outbreaks compared to purely deterministic SIR.

**State-action trajectories.** For the experiments in the main text, we generate multiple trajectories of length $T$ starting from diverse initial conditions $(S_0, I_0, R_0)$. Even though the *action* is unused in the provided snippet, we include it in the simulator's interface so that interventions (e.g., lockdowns) can be readily modeled by modifying $\beta$ on certain steps or by implementing direct changes to $(S, I, R)$. This approach follows typical RL-friendly environment design (Sutton & Barto, 2018).

**Sampling procedure for dataset generation.** To create training and evaluation datasets:

- We sample initial conditions $(S_0, I_0, R_0)$ from a broad distribution (e.g., $S_0 \sim \text{Unif}(900, 1000), I_0 \sim \text{Unif}(1, 20), R_0 = 0$).

- We simulate for $T = 60$ steps (or an alternative fixed horizon).

- We repeat this process for $N$ initial seeds, thereby obtaining $N$ state-action trajectories of length $T$.

We then split these trajectories into training, validation, and test sets (e.g., $N_{\text{train}} = 100$, $N_{\text{val}} = 100$, $N_{\text{test}} = 100$). With each trajectory, we store the transitions $(s_t, a_t, s_{t+1})$ for subsequent fitting and analysis.

**Parameter prior and bounds.** In our experiments, we typically impose a uniform prior on $\beta \in [0, 2]$ and $\gamma \in [0, 1]$, reflecting broad uncertainty over transmission and recovery rates. This range can be narrowed or broadened as needed to reflect real-world epidemiological settings.

**Key features and complexity.** Although smaller in scale than real-world COVID-19 simulators, this environment captures:

- *Stochastic transitions*: The Binomial updates ensure randomness around infection and recovery.

- *Discrete-time stepping*: Amenable to reinforcement learning or iterative policy simulation.

- *Potential policy control*: Lockdowns or other NPIs can be included by coupling the `action` with $\beta$.

This environment thereby provides a challenging testbed for building, calibrating, and evaluating simulation-based approaches—in particular, assessing how well a learned or LLM-generated model can handle partial observability and out-of-distribution interventions.

In short, our **COVID-19 SIR environment** is representative of a broader class of infectious disease models used in practice (Cooper et al., 2020; AlQadi & Bani-Yaghoub, 2022), while remaining computationally tractable for large-scale experimentation. Together with the additional environments discussed in the main paper, it forms a suite of complementary challenges evaluating the generalization, causal grounding, and empirical alignment of simulator-building approaches.

### C.2. Supply Chain Environment

Our **Supply Chain** environment is loosely inspired by the well-known "beer game" (Sterman, 1989), a classic exercise in operations research illustrating how stochastic demand and inventory pipelines can induce the "bullwhip effect" across multiple echelons of a supply chain (retailer, wholesaler, distributor, manufacturer). For simplicity, we focus here on a *single-stage retailer*, capturing key dynamics of inventory management, backlogs (unfilled demand), and delayed shipments in transit.

**State variables and partial observability.**   At each discrete time step $t$, the environment tracks:

- `inventory`: The current on-hand stock of the retailer (nonnegative integer).

- `pipeline`: A list of shipments in transit, each described by a tuple (quantity, time_remaining). Once time_remaining reaches zero, the shipment arrives in inventory.

- `backlog`: The unfilled demand from prior time steps.

- `t`: The current time index (integer).

The retailer does not directly observe future or upstream supply conditions, making this environment *partially observed*: shipments placed now can arrive with uncertain delays, reflecting real-world supply-chain complexities.

**Actions and parameters.**   The agent's action $a_t \geq 0$ indicates how many units to order at time $t$. The environment maintains four key parameters in a float array:

$$\left[\lambda_{\text{demand}}, \; c_{\text{holding}}, \; c_{\text{backlog}}, \; L_{\text{lead}}\right],$$

where:

- $\lambda_{\text{demand}}$ is the Poisson mean for random daily demand,

- $c_{\text{holding}}, \; c_{\text{backlog}}$ are (optional) costs associated with carrying inventory and having a backlog, respectively,

- $L_{\text{lead}}$ is a deterministic lead time indicating how many steps elapse before new orders arrive.

In many operational contexts, these parameters are only partially known or vary over time. In our experiments, we allow them to be fitted to real or synthetic data, reflecting the environment's calibration process.

**Core dynamics.**   The pseudo-code in Listing 2 describes how the state evolves each step:

a) **Demand sampling**: We draw a random demand $D_t$ from a Poisson distribution with mean $\lambda_{\text{demand}}$.

b) **Pipeline update**: We decrement time_remaining for all shipments in transit. Any shipment whose time_remaining $= 0$ is added to on-hand inventory.

c) **Backlog fulfillment**: If there is a backlog from previous steps, we fill as much as the current inventory allows, reducing both inventory and backlog.

d) **Current demand fulfillment**: We fill as much of the new demand $D_t$ as possible from the remaining inventory; any unfilled demand is appended to the backlog.

e) **Placing a new order**: The agent's action $a_t$ units are ordered, entering the pipeline with time_remaining $= L_{\text{lead}}$.

*Listing 2.* Step function for the single-stage supply chain environment. Orders placed at time $t$ join the pipeline with a lead time. Demand is randomly drawn from a Poisson distribution, and partial fulfillment may produce backlog.

```
class SimulatorStep:
    def __init__(self):
        # [demand_lambda, inventory_holding_cost, backlog_cost, lead_time]
        self.parameters = np.array([5.0, 1.0, 2.0, 2.0], dtype=float)
```

```
def step(self, state: dict, action: int, rng=None) -> dict:
    """
    Advance the supply chain environment by one step.
    """
    demand_lambda = self.parameters[0]
    lead_time = int(self.parameters[3])

    # 1) Random demand from Poisson
    demand = safe_poisson(demand_lambda)

    # 2) Update pipeline (shipments in transit)
    new_inventory, new_pipeline = self._update_pipeline(state)

    # 3) Fulfill backlog first
    backlog_filled = min(new_inventory, state["backlog"])
    new_inventory -= backlog_filled
    new_backlog = state["backlog"] - backlog_filled

    # 4) Fulfill today's demand
    demand_filled = min(new_inventory, demand)
    new_inventory -= demand_filled
    unsatisfied_demand = demand - demand_filled
    new_backlog += unsatisfied_demand

    # 5) Place an order (action)
    if action > 0:
        new_pipeline.append((action, lead_time))

    # 6) Build the next state
    next_state = {
        "inventory": new_inventory,
        "pipeline": new_pipeline,
        "backlog": new_backlog,
        "t": state["t"] + 1,
    }

    return next_state
...
```

**Dataset generation.** We generate state-action trajectories by simulating over a fixed horizon $T$:

- **Initial state**: We set `inventory` $\approx 20$, `pipeline` empty, `backlog` $= 0$, and `t` $= 0$.

- **Policy**: For simplicity, an agent might follow a simple reorder policy (e.g., $(s, S)$ policy or a constant order) or an $\varepsilon$-greedy approach. Alternatively, actions can be random to promote exploration.

- **Stochastic demand**: Each day, demand is drawn from $\text{Poisson}(\lambda_{\text{demand}})$.

After $N = 100$ simulated rollouts of $T = 60$, we collect $(\texttt{state}_t, \texttt{action}_t, \texttt{state}_{t+1})$ tuples to form a dataset. We then split these data into training, validation, and test sets, as described for the other environments.

**Key features and complexity.** The environment remains tractable yet exhibits hallmark properties of supply chains:

- *Partial observability*: The retailer sees only on-hand inventory and current backlog, while future shipments remain uncertain in the pipeline.

- *Delayed actions*: Orders only arrive after a deterministic lead time $L_{\text{lead}}$, echoing real-world shipping delays.

- *Stochastic demand*: Daily demand is random, requiring dynamic adjustments of orders to avoid stockouts or excessive inventory.

These elements expose strong temporal dependencies and delayed feedback loops, making it nontrivial to model or plan in. Hence, this single-stage environment is an effective proving ground for evaluating how well automatic simulator-generation methods (like G-Sim) can capture discrete transitions, uncertain arrivals, and partial observability consistent with real supply-chain operations.

### C.3. Hospital Bed Scheduling Environment

Our **Hospital Bed Scheduling** environment simulates patient admissions for three different diseases into a hospital with separate Intensive Care Unit (ICU) and standard-care ward capacities. This captures both the stochastic and discrete operational challenges often faced in healthcare settings (Green, 2006; Koizumi et al., 2005; Brailsford, 2007). Each disease has its own daily arrival rate, average length-of-stay (LOS), and mortality profile, resulting in partial observability of future admissions and bed availability.

**State variables.** We model the hospital over a discrete day-by-day timescale, where the state at day $d$ is a dictionary containing:

- `day`: The current day index (integer).

- `icu_occupancy`: Number of patients currently occupying ICU beds.

- `standard_occupancy`: Number of patients in standard-care beds.

- `patients`: A list of individual patient records, where each record includes:

    - `disease_id`: An integer $\in \{0, 1, 2\}$ identifying the disease.
    - `bed_type`: Either "ICU" or "Standard."
    - `los_remaining`: The number of days left before discharge (if the patient survives).
    - `is_alive`: A boolean indicating whether the patient is still alive.
    - `day_in_hospital`: How many days the patient has already spent hospitalized.

This structure allows us to track heterogeneous patient journeys and resource usage in detail, while also permitting partial observability (e.g., future arrivals are unknown).

**Parameters.** We store 14 parameters in a single NumPy array:

$$\underbrace{\text{arrival\_rate\_0}, \text{arrival\_rate\_1}, \text{arrival\_rate\_2}}_{\text{Poisson means for disease arrivals}},$$

$$\underbrace{\text{los\_mean\_0}, \text{los\_mean\_1}, \text{los\_mean\_2}}_{\text{mean lengths of stay for each disease}},$$

$$\underbrace{\text{base\_prob\_0}, \text{base\_prob\_1}, \text{base\_prob\_2}}_{\text{baseline mortality probabilities per day}},$$

$$\underbrace{\text{day\_factor\_0}, \text{day\_factor\_1}, \text{day\_factor\_2}}_{\text{day-based increase in mortality}},$$

$$\text{icu\_capacity}, \quad \text{standard\_capacity}.$$

These parameters govern (i) the expected arrival counts for each disease via Poisson processes, (ii) how quickly patients recover or die, and (iii) how many ICU and standard beds are available at once. In real-world hospitals, such quantities may be partially known or dynamically changing over time.

**Daily update logic.** Listing 3 provides a snippet of the day-to-day simulation. Each day:

a) **Existing patients**: For each patient, we compute a daily mortality probability based on the disease's baseline (`base_prob_i`) plus a day-dependent factor (`day_factor_i` $\times$ `day_in_hospital`). If the patient dies, we free the corresponding bed. If they survive and their LOS completes, they are discharged.

b) **New arrivals**: We sample arrivals for each disease from independent Poisson distributions and attempt to allocate them to a bed. ICU or standard-care bed assignment is disease-dependent (e.g., diseases 0/1 try ICU first, disease 2 tries standard first).

c) **Capacity constraints**: If both the ICU and standard-care wards are full, the patient is turned away (no admission).

d) **Increment time**: The simulation day index `day` is incremented by 1.

*Listing 3.* Core day-by-day update for the Hospital Bed Scheduling environment. Each day, we update existing patients (mortality/discharge) and sample new arrivals for each disease. Beds are limited by ICU and standard capacity.

```
class SimulatorStep:
    def __init__(self):
        # Example defaults: arrival rates, LOS means, mortality parameters, capacities
        self.parameters = np.array([
```

```
        1.0, 2.0, 1.5,   # arrival_rate_0 ,1,2
        5.0, 6.0, 4.0,   # los_mean_0 ,1,2
        0.01, 0.005, 0.008,  # base_prob_0 ,1,2
        0.002, 0.001, 0.0015,  # day_factor_0 ,1,2
        5.0,    # icu_capacity
        20.0    # standard_capacity
    ], dtype=float)

def step(self, state: dict, rng=None) -> dict:
    """
    Advance the simulation by one day:
      1) Update existing patients (mortality, discharge).
      2) Sample new arrivals (3 diseases).
      3) Attempt bed allocation.
      4) Increment day counter.
    """
    rng = np.random.default_rng()
    # Unpack parameters
    arr_rates = self.parameters[0:3]
    los_means = self.parameters[3:6]
    base_probs = self.parameters[6:9]
    day_factors = self.parameters[9:12]
    icu_capacity = int(5.0)
    standard_capacity = int(20.0)

    # 1) Update existing patients
    survivors = []
    for patient in state["patients"]:
        if patient["is_alive"]:
            # Calculate daily probability of death
            disease_id = patient["disease_id"]
            p_death = (base_probs[disease_id] + day_factors[disease_id] * patient["day_in_hospital"])
            p_death = np.clip(p_death, 0.0, 1.0)

            # Check if patient dies
            if rng.random() < p_death:
                patient["is_alive"] = False
                self._free_bed(state, patient["bed_type"])
            else:
                patient["los_remaining"] -= 1
                patient["day_in_hospital"] += 1
                if patient["los_remaining"] > 0:
                    survivors.append(patient)
                else:
                    self._free_bed(state, patient["bed_type"])

    state["patients"] = survivors

    # 2) Sample new arrivals
    arrivals = [rng.poisson(lam) for lam in arr_rates]

    # 3) Allocate beds
    for disease_id, num_arrivals in enumerate(arrivals):
        for _ in range(num_arrivals):
            los = max(1, int(rng.normal(los_means[disease_id], 1.0)))
            bed_type = self._try_allocate_bed(state, disease_id, icu_capacity, standard_capacity)
            if bed_type is not None:
                patient = {
                    "disease_id": disease_id, "bed_type": bed_type, "los_remaining": los,
                    "is_alive": True, "day_in_hospital": 1,
                }
                state["patients"].append(patient)

    # 4) Increment day
    state["day"] += 1

    return state
...
```

**Dataset generation.** We instantiate the simulator from an initial empty-hospital state (no patients, icu_occupancy = 0, standard_occupancy = 0, day = 0) and run it for $T$ days. By default, we do not explicitly include actions in this environment, as admissions occur automatically when a bed is available. However, one can embed a policy that, for instance, adjusts triage rules or modifies capacity expansions. We collect the resulting day-by-day trajectories of the state and produce train/validation/test splits for model calibration and evaluation.

**Key challenges.**

- *Stochastic discrete events*: Admissions, mortalities, and discharges occur in an integer, event-driven manner.

- *Capacity constraints*: If the ICU or standard ward is full, some patients cannot be admitted, which leads to censored or denied admissions.

- *Disease heterogeneity*: Each disease has a unique arrival rate and mortality profile, increasing the complexity of cross-disease interactions (e.g., competition for ICU beds).

- *Partial observability*: Future arrivals and disease trajectories are unknown, and patient-level details evolve stochastically day by day.

Thus, this environment approximates essential features of inpatient hospital management for multiple disease conditions, making it a valuable testbed for assessing how well automatic simulators (such as those generated by G-Sim) can handle real-world scheduling and resource-allocation dynamics (Green, 2006; Koizumi et al., 2005; Brailsford, 2007).

### C.4. Performing Intervention Insight Experiments

Our insight experiments (Section 6.1) evaluate a simulator's ability to handle *what if?* questions, particularly those involving interventions that alter the underlying system dynamics—a key requirement for **(P0) System-wide Experimentation**. These interventions often go beyond simply changing actions within the historical data distribution; they involve modifying the environment's parameters or structure.

**How Interventions are Implemented.** To perform these experiments, we follow a two-step process:

1. **Ground Truth Modification:** We first identify the specific parameter or component within the ground truth simulator's code that corresponds to the desired intervention. For example, in the COVID-19 lockdown experiment, we directly access the $\beta$ (infection rate) parameter and scale it by a multiplier $\alpha$ during the specified lockdown period. Similarly, for supply chain optimization, we modify parameters like $\ell$ (lead time) or add extra capacity $\Delta C$.

2. **G-Sim Modification:** We then inspect the final simulator code generated and calibrated by G-Sim. Thanks to its LLM-driven, often modular and interpretable structure, we can identify the parameter or code section analogous to the one modified in the ground truth. For instance, we locate G-Sim's infection rate parameter and apply the same scaling factor $\alpha$.

**Why Other Methods Fall Short.** Performing such structural or parametric interventions is generally not feasible with the other baseline methods evaluated:

- **Data-Driven World Models (RNN, Transformer, DyNODE):** These models learn a black-box mapping from states and actions to next states. They lack explicitly defined, interpretable parameters (like $\beta$ or $\ell$) that can be directly modified to reflect a change in the environment's fundamental dynamics. Intervening would require retraining on new data reflecting the change, which defeats the purpose of "what if" analysis on unseen scenarios.

- **Equation Discovery (SINDy, Genetic Program):** While these methods aim to find equations, the resulting models are often monolithic and may not expose parameters that directly correspond to meaningful, real-world interventions like adding beds or changing supply lead times. Their focus is on fitting observed dynamics, not creating an intervenable representation.

# D. Implementation Details for Baseline Methods

We provide below the implementation details of all baseline methods from the main paper. Unless otherwise noted, each baseline uses the same dataset splits (training, validation, test) and is trained to predict one-step-ahead transitions from $(s_t, a_t)$ to $s_{t+1}$. We tune hyperparameters on the validation set and apply early stopping with a patience of 20 epochs. Unless otherwise stated, models are trained for up to 2,000 epochs using the Adam (Kingma & Ba, 2014) optimizer.

**DyNODE**  *DyNODE* (Chen et al., 2018; Alvarez et al., 2020) extends neural ODEs by incorporating control actions explicitly. Specifically, we let

$$\frac{d\mathbf{z}(t)}{dt} = f\big(\mathbf{z}(t), \mathbf{u}(t); \theta\big),$$

where $\mathbf{z}(t)$ is a latent state encoding the environment, and $\mathbf{u}(t)$ is the action. To implement DyNODE as a one-step predictor, we solve this ODE numerically over each time interval $[t, t+1]$. Concretely, we use a 3-layer MLP with 128 hidden units per layer and `tanh` activations to represent $f$. We initialize weights with Xavier initialization and apply the Adam optimizer with a learning rate of $10^{-2}$. We set a batch size of 1,000 and stop if validation loss does not improve after 20 epochs.

**Transformer**  *Transformer*, from Causal Transformer (Melnychuk et al., 2022), is a modern sequence model (Vaswani et al., 2017) that can handle long-range dependencies in time series. We flatten the state-action pairs over time and feed them into a single Transformer encoder. Our implementation uses:

- An embedding dimension of 250, learned via a linear projection from the input state-action dimension.

- A single Transformer encoder block, with multi-head attention (10 heads), hidden dimension 250, and dropout probability 0.1.

- A final linear layer mapping the encoder output to the next-state prediction.

We train using AdamW (Loshchilov et al., 2017) at a learning rate of $5 \times 10^{-5}$ with a step learning-rate scheduler (step size 1.0, gamma 0.95), gradient clipping at 0.7, and a batch size of 1,000. As with DyNODE, training continues for up to 2,000 epochs or until early stopping triggers.

**RNN**  *RNN* (Rumelhart et al., 1986) is a classical recurrent neural network for autoregressive time-series modeling. We adopt a two-layer Gated Recurrent Unit (GRU) architecture, each layer having a hidden size of 250. The model takes as input the state-action vector at each time step and outputs the predicted next state. We train with the same Adam configuration as DyNODE (learning rate $10^{-2}$, batch size 1,000, 2,000 max epochs, early stopping patience of 20). All input and output features are normalized by statistics derived from the training split.

**SINDy**  *Sparse Identification of Nonlinear Dynamics* (SINDy) (Brunton et al., 2016) attempts to discover closed-form equations directly from time-series data. After estimating derivatives via finite differences, SINDy performs sparse regression over a predefined library of candidate functions (e.g., polynomials) to identify a few terms that best explain the data. We use a polynomial library of order 2, i.e. $\{1, x_i, x_i x_j, \dots\}$, and set the regularization factor $\alpha = 0.5$. We threshold small coefficients below $0.02$ to enforce sparsity for most tasks; for particularly noisy or large-scale tasks, we tune the threshold (e.g., for the COVID-19 environment). This yields a symbolic equation per dimension of the state.

**Genetic Program**  *Genetic Program* is a symbolic regression approach that searches for analytical expressions describing the environment dynamics via evolutionary algorithms. We use the implementation of Deap (De Rainville et al., 2012). We maintain a population of candidate expressions, iteratively mutating and recombining them, selecting those with the lowest mean-squared error on the training set. We constrain expressions to a predefined set of operators (e.g., $\{+, -, \times, \div\}$) and elementary functions (e.g., polynomials, exponentials). We run for a maximum of 1,000 generations with a population size of 500 and a mutation rate of 0.1.

**Ablations of G-Sim**  We also include two ablations of our proposed G-Sim pipeline, specifically for the G-Sim – ES variant:

(a) **G-Sim-ES Abl. ZeroShot**: Uses the LLM to generate code for the simulator structure once (without subsequent refinements) and does not optimize any parameters. This illustrates the performance of naive, uncalibrated LLM outputs.

(b) **G-Sim-ES Abl. ZeroShotOptim**: Uses the same LLM-generated simulator structure as (a) but applies a gradient-free optimizer to tune numerical parameters only. In contrast to full G-Sim, no structural revisions are performed.

**Shared training protocol.** All methods rely on the same one-step prediction loss, typically mean-squared error. Table 2 in the main text summarizes their performance across our benchmark environments.

# E. G-Sim Implementation Details

In the following we detail the full methodology for G-Sim, including pseudocode, training procedures, prompt templates, and diagnostics-driven refinement. Our approach builds on the framework described in Section 3 of the main paper.

## E.1. Overall G-Sim Framework

Recall that G-Sim comprises three main components:

- **LLM-Driven Structural Proposals** (Section 3, §3.1),
- **Flexible Parameter Calibration** via a choice of GFO or SBI (Section 3, §3.2),
- **Diagnostics-Driven Iterative Refinement** (Section 3, §3.3).

Each iteration produces a candidate simulator with an estimated parameter set, evaluates it via diagnostics, and refines it. Algorithm 1 gives pseudocode for the overall loop.

## E.2. Pseudocode

We provide detailed pseudocode for the G-Sim construction in Algorithm 1. (1) We maintain a history of candidate simulators for reference. (2) We prompt the LLM to generate (or refine) a structural configuration $\lambda$, including submodules and couplings. (3) We perform calibration using GFO or SBI to find its numerical parameters $\omega$. (4) We compute diagnostic scores; if improvements are needed, textual feedback is compiled for the LLM to guide a refined structural design on the next iteration. (5) We track and eventually return the best simulator found.

---

**Algorithm 1** G-Sim: High-Level Pseudocode

---

**Require:**
- Domain knowledge $\mathcal{K}$ (text descriptions, constraints),
- Training data $\mathcal{D} = \{\mathcal{D}^{(1)}, \ldots, \mathcal{D}^{(L)}\}$,
- LLM with a prompt function PromptLLM$(\cdot)$,
- Calibration engine CalibrateParams$(\cdot)$ (either GFO or SBI),
- Diagnostics function Diag$(\lambda, \omega; \mathcal{D})$,
- Maximum iterations $G$, patience for early stopping.

**Ensure:** A fully calibrated simulator $(\lambda^*, \omega^*)$ minimizing the diagnostic score.

1: Initialize History $\leftarrow \varnothing$
2: **for** $g = 1$ to $G$ **do**
3:     **(A) LLM structural proposal:**
4:       $\lambda^{(g)} \leftarrow$ PromptLLM(History, $\mathcal{K}$)
5:     **(B) Parameter calibration:**
6:       $\omega^{(g)} \leftarrow$ CalibrateParams$(\lambda^{(g)}, \mathcal{D})$    // Using either GFO or SBI
7:     **(C) Diagnostics:**
8:       $d^{(g)} =$ Diag$(\lambda^{(g)}, \omega^{(g)}; \mathcal{D})$
9:     **(D) Record current candidate:**
10:       History $\leftarrow$ History $\cup \{(\lambda^{(g)}, \omega^{(g)}, d^{(g)})\}$
11:     **(E) Refinement check:**
12:     **if** early stopping criterion met (e.g., no improvement for $p$ iterations) **then**
13:       **break**
14:     **else**
15:       **Compile textual feedback** $\psi^{(g)}$ from diagnostics
16:       **Append** $\psi^{(g)}$ **to prompt context** for next iteration
17:     **end if**
18: **end for**
19: **Return** $(\lambda^*, \omega^*)$ with best (lowest) $d^{(g)}$ in History.

---

## E.3. Training and Calibration Details

Once the LLM provides a structural design $\lambda$, we must *calibrate* its numerical parameters $\omega$. These parameters can be real-valued or discrete. We employ two primary methods: Gradient-Free Optimization (GFO) and Simulation-Based Inference (SBI).

### E.3.1. GRADIENT-FREE OPTIMIZATION (GFO) USING EVOLUTIONARY STRATEGIES (ES)

GFO provides a robust method for parameter fitting, especially for simulators that are non-differentiable, stochastic, or involve discrete parameters.

**Fitness/Objective.** The fitness function $\mathcal{J}(\omega, \lambda)$ is the **Mean Squared Error (MSE)** between the simulator's predicted outputs and the ground-truth observations from $\mathcal{D}$. The GFO search aims to find $\omega^* = \arg\min_\omega \mathcal{J}(\omega, \lambda)$.

**Implementation with EvoTorch.** We implement the GFO step using the *GeneticAlgorithm* class from *EvoTorch*. The process involves:

1. **Initialization**: A population of candidate parameter sets $\{\omega_i\}$ is initialized.
2. **Evaluation**: Each candidate $\omega_i$ is evaluated by running the simulator $F(\cdot; \lambda, \omega_i)$ and computing its fitness (MSE).
3. **Selection**: Tournament selection chooses individuals for the next generation.
4. **Variation**: `SimulatedBinaryCrossOver` and `GaussianMutation` create new candidates.
5. **Iteration**: Steps 2-4 repeat across generations.

Our key EvoTorch settings are:

- *Population size*: 200,
- *Number of generations*: 10,
- *Search operators*:
    - `SimulatedBinaryCrossOver` with tournament size 4, crossover rate 1.0, and $\eta = 8$,
    - `GaussianMutation` with standard deviation `stdev=0.03`.

This population-based approach offers resilience to local minima. We also *warm start* calibration from the best parameters found previously when only the structure changes, balancing exploration and refinement.

### E.3.2. SIMULATION-BASED INFERENCE (SBI)

As an alternative, we use SBI for Bayesian parameter inference. This is particularly useful when uncertainty quantification is desired.

**Simulation Budget.** Based on our implementation, we use a simulation budget of *1,000 simulations* to train the SBI posterior estimator.

**Inference and Posterior Estimation.** We use the Neural Posterior Estimation (NPE) algorithm from the 'sbi' library. The simulator's output, a trajectory of states, is flattened into a single vector to serve as the observation for the density estimator.

1. **Simulate**: We draw parameters from a uniform prior defined by the simulator's 'get_parameters_uniform_prior_min_max' method and run the simulator to generate pairs of parameters and observation vectors $(\theta, x)$.
2. **Train**: We train a neural density estimator (e.g., a Masked Autoregressive Flow) on these pairs to approximate the posterior $p(\omega|\mathcal{D})$.
3. **Sample and Estimate**: We sample from the learned posterior to obtain a distribution of plausible parameters. For evaluation and generating the final simulator, we use the **mean of the posterior samples** as the point estimate for $\omega^*$.

This process provides not only a point estimate but also allows for the full posterior to be inspected for uncertainty analysis.

## E.4. Prompt Templates and Structural Generation

The LLM-based structural proposals rely on a set of *prompt templates* that describe the domain context, submodule templates, coupling schemes, and known constraints. Specifically, for all LLM experiments we used the O1 model from OpenAI. We show the general structure below; these are typically fed as `system` or `user` messages in an API like OpenAI or other LLM interfaces:

### System Prompt

```
Objective: Write code to create an accurate and realistic simulator for a given task in NumPy.
Please note that the code should be fully functional. No placeholders.

You must act autonomously and you will receive no human input at any stage. You have to return as output the complete code for completing this task, and correctly
    improve the code to create the most accurate and realistic simulator possible.
You always write out the code contents. You always indent code with tabs.
You cannot visualize any graphical output. You exist within a machine. The code can include black box multi-layer perceptions where required.

Use the functions provided. When calling any helper function, only provide a RFC8259 compliant JSON request (no additional text or formatting).
```

### Main Prompt (followed by system prompt for the COVID-19 task)

```
You will get a simulator description to code a **`step` function** in NumPy.

System Description:
```
COVID SIR environment.

Here you must model the simulation step with the below state and action of

The environment state is represented by a dictionary:
"S": int, "I": int, "R": int

Action: None = None

The collected trajectory lasts for 60 time steps (days).
```

Modelling goals:```
* The parameters of the simulator will be optimized to an observed training state-action dataset with the given simulator using simulation-based Inference.
* The observed training dataset has very few samples, and the model must be able to generalize to unseen state-action data.
```

Requirement Specification:```
* The code generated should achieve the lowest possible validation Wasserstein loss, of 1e-10 or less.
* The code generated should be interpretable, and fit the dataset as accurately as possible.
```

Skeleton code to fill in:```
class SimulatorStep():
    def __init__(self):
        # TODO: Fill in the code here - define the parameters of the model and make them self here.

    def step(self, state: dict, action: int, rng: np.random.Generator) -> dict:
        # Must include all the logic
        ...
        return next_state

    def get_parameters(self) -> np.ndarray:
        """
        Returns the model parameters as an array.
        """
        # TODO: Fill in the code here

    def set_parameters(self, parameters: np.ndarray):
        """
        Updates the model parameters.

        Args:
            parameters (np.ndarray): Array of parameters to update.
        """
        # TODO: Fill in the code here

    def get_parameters_uniform_prior_min_max -> np.ndarray:
        """
        Returns the uniform prior bounds for the parameters.

        Returns:
            np.ndarray: Array of shape (2, num_parameters) with min and max bounds.
        """
        # TODO: Fill in the code here
```

Useful to know:
```
* The generated code must include the complete `step` function body in NumPy, fully functional, no placeholders.
* You are a code evolving machine, and you will be called 20 times to generate code, and improve the code to achieve the lowest possible validation Wasserstein
    loss.
* The model defines the possibly stochastic transition function taking the full state, action and predicting the next state, and will be used to fit the observed
    training dataset.
* You can use any parameters you want however, you have to define these.
* It is preferable to decompose the system into compartments if possible.
* You can use any unary functions, for example log, exp, power etc.
* You can use numpy sampling distributions.
* Under no circumstance can you change the skeleton code function definitions, only fill in the code.
* Make sure your code follows the exact code skeleton specification.
* When defining categorical distributions that are parameterized make it so that the probabilities are automatically normalized as they will be sampled as random
    values. I.e. normalize the probabilities within the step function.
```

Think step-by-step, and then give the complete full working code. You are generating code for iteration 0 out of 5.
```

At iteration $g$, we also append a short textual *feedback summary* describing any mismatches or domain rules that the previous design missed. For instance:

1) Consider including an additional parameter that scales the binomial trials' variance. If the dataset's transitions have more or less variability than pure binomial draws, adding a dispersion parameter and adjusting the sampling mechanism accordingly can improve fit.

2) Evaluate using a more direct approach for probability of infection and recovery. For instance, instead of relying on $(1 - \exp(-\beta \times I / N))$ and $(1 - \exp(-\gamma))$, you could compute probabilities directly or use a saturating function (e.g., a logistic) if the data suggest better alignment with time-dependent usage.

3) Observe that the optimized $\beta$ and $\gamma$ values (about 0.4486 and 0.0841) might not fully capture the infection/recovery process if the real dataset implies changing contact rates over time. You can allow for a time-dependent or state-dependent factor (another parameter) to refine the infection rate or to capture partial immunity effects.

4) Include a small "immunity loss" or "reinfection" parameter if the real data show recovered individuals occasionally re-entering the infected compartment. This can be done by adding an additional parameter controlling the probability of moving from R back to S.

5) If the dataset's scale is large, consider adding a parameter for underreporting or unobserved asymptomatic infections. In effect, you use a hidden compartment from which infected are not directly observed, adjusting the probabilities in the step function to better match observed transitions.

6) Keep the prior bounds in get_parameters_uniform_prior_min_max sufficiently flexible to allow exploration of new parameters (e.g., multiple parameters if you add reinfection, underreporting, etc.). This will help ensure your inference procedure can discover better-fitting solutions.

By refining these aspects, you can improve accuracy and realism and potentially lower the validation Wasserstein loss further.

## This crucially arises from this reflection prompt

Please reflect on how you can improve the code to fit the dataset as accurately as possible, and be realistic. Think step-by-step. Provide only actionable feedback, that has direct changes to the code. Do not write out the code, only describe how it can be improved. Where applicable use the values of the optimized parameters to reason how the code can be improved to fit the dataset as accurately as possible. This is for generating new code for the next iteration 1 out of 5.

Importantly this is appended after a prompt that includes all previous models, their losses, and their parameters optimized to the observed and validation dataset. Here is an example from the COVID-19 task.

You generated the following code completions, which then had their parameters optimized to the training dataset. Please reflect on how you can improve the code to minimize the validation loss to 1e−6 or less. The code examples are delineated by ###.

Here are your previous iterations the best programs generated.'''
Iteration 0. Best Val Loss: {'mse': 2.842185792349727, 'mmd': 0.0041523609029386396, 'wass': 0.41401221629537344}. Model description: This code implements a discrete SIR model in NumPy for COVID progression, parameterized by infection rate \beta and recovery rate \gamma. We incorporate stochastic transitions via binomial sampling. The get/set methods allow for parameter optimization, while the step method updates the number of susceptible, infected, and recovered individuals realistically. By ensuring a flexible range for \beta and \gamma, we support better fitting of the data and minimize the Wasserstein loss. This approach should enable effective simulation-based inference.
Iteration 1. Best Val Loss: {'mse': 2.842185792349727, 'mmd': 0.0041523609029386396, 'wass': 0.41401221629537344}. Model description: This code implements a discrete SIR model in NumPy for COVID progression, parameterized by infection rate \beta and recovery rate \gamma. We incorporate stochastic transitions via binomial sampling. The get/set methods allow for parameter optimization, while the step method updates the number of susceptible, infected, and recovered individuals realistically. By ensuring a flexible range for \beta and \gamma, we support better fitting of the data and minimize the Wasserstein loss. This approach should enable effective simulation-based inference.
'''

Here are the top code completions so far that you have generated, sorted for the lowest validation loss last:'''

Val Wasserstein Loss: 0.474 (Where the val MSE loss per dimension is susceptible val MSE loss: 2.12, infected val MSE loss: 4.09, recovered val MSE loss: 2.1)
    Iteration: 1
###
'''
import numpy as np

class SimulatorStep():
        def __init__(self):
                # Define parameters: [beta, gamma, omega]
                # Default values, can be changed upon parameter tuning
                self.params = np.array([0.45, 0.08, 0.01], dtype=float)

        def step(self, state: dict, action: int, rng: np.random.Generator) -> dict:
                S, I, R = state['S'], state['I'], state['R']
                beta, gamma, omega = self.params[0], self.params[1], self.params[2]
                N = S + I + R
                if N <= 0:
                        return {"S": S, "I": I, "R": R}
                prob_infection = 1.0 - np.exp(-beta * ((I + 1.0) / N)) if I >= 0 else 0.0
                prob_infection = max(0.0, min(prob_infection, 1.0))
                new_infections = rng.binomial(S, prob_infection)
                prob_recovery = 1.0 - np.exp(-gamma)
                prob_recovery = max(0.0, min(prob_recovery, 1.0))
                new_recoveries = rng.binomial(I, prob_recovery)
                prob_reinfection = 1.0 - np.exp(-omega)
                prob_reinfection = max(0.0, min(prob_reinfection, 1.0))
                new_reinfections = rng.binomial(R, prob_reinfection)
                S_next = S - new_infections + new_reinfections
                I_next = I + new_infections - new_recoveries
                R_next = R + new_recoveries - new_reinfections
                next_state = {"S": S_next, "I": I_next, "R": R_next}
                return next_state
        def get_parameters(self) -> np.ndarray: return self.params.copy()
        def set_parameters(self, parameters: np.ndarray): self.params = parameters.astype(float)
        def get_parameters_uniform_prior_min_max(self) -> np.ndarray:
                return np.array([[0.0, 0.0, 0.0], [1.0, 1.0, 0.2]])
'''
optimized_parameters = [0.4106612801551819, 0.082090824842453, -0.01027984619140625]
###

Val Wasserstein Loss: 0.414 (Where the val MSE loss per dimension is susceptible val MSE loss: 2.18, infected val MSE loss: 4.32, recovered val MSE loss: 2.23)
    Iteration: 0
###
'''
import numpy as np

```
class SimulatorStep():
    def __init__(self):
        self.params = np.array([0.1, 0.05], dtype=float)
    def step(self, state: dict, action: int, rng: np.random.Generator) -> dict:
        S, I, R = state['S'], state['I'], state['R']
        beta, gamma = self.params[0], self.params[1]
        N = S + I + R
        if N <= 0: return {"S": S, "I": I, "R": R}
        prob_infection = 1.0 - np.exp(-beta * (I / N)) if I > 0 else 0.0
        new_infections = rng.binomial(S, max(0.0, min(prob_infection, 1.0)))
        prob_recovery = 1.0 - np.exp(-gamma)
        new_recoveries = rng.binomial(I, max(0.0, min(prob_recovery, 1.0)))
        S_next = S - new_infections
        I_next = I + new_infections - new_recoveries
        R_next = R + new_recoveries
        return {"S": S_next, "I": I_next, "R": R_next}
    def get_parameters(self) -> np.ndarray: return self.params.copy()
    def set_parameters(self, parameters: np.ndarray): self.params = parameters.astype(float)
    def get_parameters_uniform_prior_min_max(self) -> np.ndarray:
        return np.array([[0.0, 0.0], [1.0, 1.0]])
```
optimized_parameters = [0.448597252368927, 0.08406898379325867]
###
```

Please reflect on how you can improve the code to fit the dataset as accurately as possible, and be realistic. Think step-by-step. Provide only actionable feedback, that has direct changes to the code. Do not write out the code, only describe how it can be improved. Where applicable use the values of the optimized parameters to reason how the code can be improved to fit the dataset as accurately as possible. This is for generating new code for the next iteration 2 out of 5.

This feedback helps the LLM propose refined structures in the next iteration.

### E.5. Diagnostics Computation and Refinement

**Diagnostic Measures.** At the end of each iteration, we compute two main types of diagnostic metrics:

- *Predictive Discrepancy*: $\delta_{\text{predictive}}$, e.g., MSE or Wasserstein distances between real and simulated trajectories on a validation set.
- *Domain Violations*: $\delta_{\text{domain}}$, e.g., negative capacities, ignoring constraints, or failing plausibility checks such as "transfers must not exceed the available capacity in the previous step."

The first is computed quantitatively, while the latter is evaluated qualitatively using a reflection prompt that interprets the previously generated and optimized code structure model, along with its optimized parameters.

In our practical implementation, the quantitative predictive discrepancy is key. We compute both Mean Squared Error (MSE) and the 1-Wasserstein distance ($W_1$) between the simulated outputs and a held-out validation dataset. The $W_1$ distance serves as the primary fitness score used for ranking different simulator structures and for the early stopping criterion.

However, for generating textual feedback for the LLM's reflection step, we provide a more detailed breakdown. We compute the MSE for each individual output dimension of the simulator. By presenting both the overall $W_1$ score and this per-dimension MSE breakdown to the LLM, we enable it to reason about *which* specific components are causing the largest errors and thus require structural refinement in the next iteration.

**Refinement.** Based on these quantitative (MSE, $W_1$) and qualitative (reflection prompt output) diagnostics, we compile a textual summary $\psi^{(g)}$ enumerating the major shortfalls. This textual message is appended to the next LLM prompt (§E.4), guiding the LLM to propose a revised $\lambda^{(g+1)}$ to address these shortcomings. This iterative loop is conceptually illustrated in Figure 1.

### E.6. Implementation Notes

**Parallel vs. Iterative LLM Calls.** Depending on the LLM service, we can prompt it once per iteration in a purely *serial* manner or maintain a short conversation chain. We typically keep a short conversation for each design iteration: (1) provide domain $\mathcal{K}$, (2) incorporate textual feedback from the last iteration, (3) ask the LLM for a revised structural design. This "reflective" approach is simpler to implement in practice than a single mega-prompt, and it helps the LLM keep track of incremental changes.

**Hyperparameters for G-Sim.** In our experiments, we typically use a maximum of 5 refinement loops, a patience of 3 for early stopping, a population size of 200 in evolutionary search, 10 generations, and a mutation rate of 0.03 for parameter changes.

### E.7. Full G-Sim Workflow Summary

Putting it all together:

1. **Initialize** an empty history $\text{History}$.
2. **LLM Proposes Structure**:
   (a) Use domain knowledge $\mathcal{K}$ to propose or refine $\lambda$.
   (b) Incorporate textual feedback from prior attempts.
3. **Calibrate Parameters** $\omega$ via GFO or SBI:
   (a) Simulate $F(\cdot; \lambda, \omega)$ and compare with real data $\mathcal{D}$.
   (b) Update $\omega$ to optimize the chosen objective.
4. **Compute Diagnostics**:
   (a) Evaluate predictive mismatch $\delta_{\text{predictive}}$,
   (b) Check domain constraints $\delta_{\text{domain}}$,
   (c) Combine into $d^{(g)}$.
5. **Refine or Stop**:
   (a) If stopping criterion is met, stop and return the best simulator found.
   (b) Otherwise, compile textual feedback $\psi^{(g)}$ and *goto* Step 2.

This iterative loop typically converges to a simulator that balances **(i)** plausible structural forms aligned with domain knowledge, and **(ii)** accurate parameter estimates aligned with empirical data. In Section 5 of the main paper, we show empirical examples of G-Sim performance on real-world-inspired tasks.

### E.8. Implementation of the Diagnostic Function $\text{Diag}$

Below is a more explicit representation of how we combine submodule-level error terms and domain rule violations into a single diagnostic score:

$$d(\lambda, \omega) \;=\; \sum_{k=1}^{K} w_k \, \text{Err}^k(\mathcal{D}^{(k)}, F^k(\cdot; \omega^k, \lambda))$$

Here, $\text{Err}^k$ measures the discrepancy for the $k$-th submodule's partial observation $\mathcal{D}^{(k)}$. This is implemented as validation MSE per output dimension to enable the LLM to reason over which component is incorrectly specified and by how much to improve upon in the next iteration.

### E.9. Observed Speed-Ups via Parallelization

When G-Sim is implemented with a population-based evolutionary algorithm, the majority of computation in each iteration is devoted to the repeated simulation over population members. We found that distributed parallelization across multiple CPU or GPU workers yields near-linear speed-ups for the tasks in Section 5, thus making G-Sim practical even for more complex submodule designs.

### E.10. Handling Stochastic Simulators

In domains like queueing or epidemiology, submodules produce random transitions. Our approach simply draws multiple sample trajectories to evaluate the expected mismatch for each candidate $\omega$. In practice, we set a small number of Monte Carlo draws (200) to keep the computational overhead manageable. The same approach extends to partial data or events (e.g., times-to-event) via likelihood-based scoring.

### E.11. Summary of Key Implementation Steps

1. **Give domain constraints and partial data coverage details** to guide feasible structures.

2. **Initialize parameter search** using either GFO/ES or SBI, parallelizing as feasible.

3. **Refine structure** via textual feedback if domain constraints or high predictive errors are uncovered.

4. **Stop** when the combined diagnostic meets the threshold or the maximum iteration limit is reached.

The resulting simulator is then used to run "what if?" analyses or further submodule-level modifications as needed.

## F. Illustrative G-Sim Prompt Example

Below is an example prompt of text we provide to the LLM to generate a supply chain simulation:

```
System Description:
```
Single-stage Beer Game supply chain environment.

Here you must model the simulation step with the below state and action of

The environment state is represented by a dictionary:
    "inventory": int,            # On-hand units
    "pipeline": list of (int, int),  # shipments: (quantity, time_remaining)
    "backlog": int,              # units of unfilled demand
    "t": int                     # current time step

Action: int = is the new order to place with the supplier. That creates a pipeline entry with a lead time sampled from a discrete distribution.

The collected trajectory lasts for 60 time steps (days).
```
Modelling goals:```
* The parameters of the simulator will be optimized to an observed training state-action dataset with the given simulator using simulation-based Inference.
* The observed training dataset has very few samples, and the model must be able to generalize to unseen state-action data.
```
Requirement Specification:```
* The code generated should achieve the lowest possible validation Wasserstein loss, of 1e-10 or less.
* The code generated should be interpretable, and fit the dataset as accurately as possible.
```
Skeleton code to fill in:```
class SimulatorStep():
    def __init__(self):
        # TODO: Fill in the code here - define the parameters of the model and make them self here.

    def step(self, state: dict, action: int, rng: np.random.Generator) -> dict:
        # Must include all the logic
        ...
        return next_state

    def get_parameters(self) -> np.ndarray:
        """
        Returns the model parameters as an array.
        """
        # TODO: Fill in the code here

    def set_parameters(self, parameters: np.ndarray):
        """
        Updates the model parameters.

        Args:
            parameters (np.ndarray): Array of parameters to update.
        """
        # TODO: Fill in the code here

    def get_parameters_uniform_prior_min_max -> np.ndarray:
        """
        Returns the uniform prior bounds for the parameters.

        Returns:
            np.ndarray: Array of shape (2, num_parameters) with min and max bounds.
        """
        # TODO: Fill in the code here
```

Useful to know:
```
* The generated code must include the complete `step` function body in NumPy, fully functional, no placeholders.
* You are a code evolving machine, and you will be called 20 times to generate code, and improve the code to achieve the lowest possible validation Wasserstein loss.
* The model defines the possibly stochastic transition function taking the full state, action and predicting the next state, and will be used to fit the observed training dataset.
* You can use any parameters you want however, you have to define these.
* It is preferable to decompose the system into compartments if possible.
* You can use any unary functions, for example log, exp, power etc.
* You can use numpy sampling distributions.
* Under no circumstance can you change the skeleton code function definitions, only fill in the code.
* Make sure your code follows the exact code skeleton specification.
* When defining categorical distributions that are parameterized make it so that the probabilities are automatically normalized as they will be sampled as random values. I.e. normalize the probabilities within the step function.
```

Think step-by-step, and then give the complete full working code. You are generating code for iteration 0 out of 5.
```

The LLM's response typically includes:

- A set of submodules (Python classes or code blocks),

- Proposed couplings (i.e., how arrivals feed into the occupancy submodule),

- Hard-coded initial values for $\omega$ (which the calibration then refines).

If the simulator violates domain constraints, we compile feedback for the next iteration.

## F.1. Env Prompts

We provide the environment prompts per environment in the previous examples. For completion we also include the Hospital Bed Scheduling prompt below.

```
You will get a simulator description to code a **`step` function** in NumPy.

System Description:
```
**Three-disease Hospital Environment**

This environment simulates a hospital with separate **ICU** and **standard** beds, along with patients from **3 different disease types**.

### State
Represented by a dictionary:
    - **"day"** *(int)*
        The current simulation day (time step).
    - **"icu_occupancy"** *(int)*
        Number of **ICU** beds currently occupied.
    - **"standard_occupancy"** *(int)*
        Number of **standard** (non-ICU) beds currently occupied.
    - **"patients"** *(list of dicts)*
        Each patient dictionary has:
        - **"disease_id"**: An integer \(\{0, 1, 2\}\) identifying the disease type.
        - **"bed_type"**: Either `"ICU"` or `"Standard"`.
        - **"los_remaining"** *(int)*: Remaining length of stay in days.
        - **"is_alive"** *(bool)*: Whether the patient is still alive.
        - **"day_in_hospital"** *(int)*: How many days the patient has been in the hospital.

### Action
There is no direct external "action" to take each day. Instead:
1. New patients arrive
2. The environment attempts to **allocate** each new arrival to an **ICU** or **standard** bed according to predefined rules and capacities.
3. If no suitable bed is available, the patient is not admitted.

### Episode Length
A typical simulation might run for a fixed number of days, for example **60** time steps, after which the simulation ends.
```

Modelling goals:```
* The parameters of the simulator will be optimized to an observed training state-action dataset with the given simulator using simulation-based Inference.
* The observed training dataset has very few samples, and the model must be able to generalize to unseen state-action data.
```

Requirement Specification:```
* The code generated should achieve the lowest possible validation Wasserstein loss, of 1e-10 or less.
* The code generated should be interpretable, and fit the dataset as accurately as possible.
```

Skeleton code to fill in:```
class SimulatorStep():
    def __init__(self):
        # TODO: Fill in the code here - define the parameters of the model and make them self here.

    def step(self, state: dict, rng: np.random.Generator) -> dict:
        # Must include all the logic
        ...
        return next_state

    def get_parameters(self) -> np.ndarray:
        """
        Returns the model parameters as an array.
        """
        # TODO: Fill in the code here

    def set_parameters(self, parameters: np.ndarray):
        """
        Updates the model parameters.

        Args:
            parameters (np.ndarray): Array of parameters to update.
        """
        # TODO: Fill in the code here

    def get_parameters_uniform_prior_min_max -> np.ndarray:
        """
        Returns the uniform prior bounds for the parameters.

        Returns:
            np.ndarray: Array of shape (2, num_parameters) with min and max bounds.
        """
        # TODO: Fill in the code here
```

Useful to know:
```
* The generated code must include the complete `step` function body in NumPy, fully functional, no placeholders.
* You are a code evolving machine, and you will be called 20 times to generate code, and improve the code to achieve the lowest possible validation Wasserstein loss.
* The model defines the possibly stochastic transition function taking the full state, action and predicting the next state, and will be used to fit the observed training dataset.
* You can use any parameters you want however, you have to define these.
* It is preferable to decompose the system into compartments if possible.
* You can use any unary functions, for example log, exp, power etc.
* You can use numpy sampling distributions.
* Under no circumstance can you change the skeleton code function definitions, only fill in the code.
* Make sure your code follows the exact code skeleton specification.
* When defining categorical distributions that are parameterized make it so that the probabilities are automatically normalized as they will be sampled as random values. I.e. normalize the probabilities within the step function.
* Do not include any action input in the step function, as there is no action input. Always explicitly follow the skeleton code function definitions, you are not allowed to change them.
```

Think step-by-step, and then give the complete full working code. You are generating code for iteration 0 out of 5.
```

## COVID-19

You will get a simulator description to code a **`step` function** in NumPy.

System Description:
```
COVID SIR environment.

Here you must model the simulation step with the below state and action of

The environment state is represented by a dictionary:
"S": int, "I": int, "R": int

Action: None = None

The collected trajectory lasts for 60 time steps (days).
```

Modelling goals:```
* The parameters of the simulator will be optimized to an observed training state-action dataset with the given simulator using simulation-based Inference.
* The observed training dataset has very few samples, and the model must be able to generalize to unseen state-action data.
```

Requirement Specification:```
* The code generated should achieve the lowest possible validation Wasserstein loss, of 1e-10 or less.
* The code generated should be interpretable, and fit the dataset as accurately as possible.
```

Skeleton code to fill in:```
```
class SimulatorStep():
    def __init__(self):
        # TODO: Fill in the code here - define the parameters of the model and make them self here.

    def step(self, state: dict, action: int, rng: np.random.Generator) -> dict:
        # Must include all the logic
        ...
        return next_state

    def get_parameters(self) -> np.ndarray:
        """
        Returns the model parameters as an array.
        """
        # TODO: Fill in the code here

    def set_parameters(self, parameters: np.ndarray):
        """
        Updates the model parameters.

        Args:
            parameters (np.ndarray): Array of parameters to update.
        """
        # TODO: Fill in the code here

    def get_parameters_uniform_prior_min_max -> np.ndarray:
        """
        Returns the uniform prior bounds for the parameters.

        Returns:
            np.ndarray: Array of shape (2, num_parameters) with min and max bounds.
        """
        # TODO: Fill in the code here
```
```

Useful to know:
```
* The generated code must include the complete `step` function body in NumPy, fully functional, no placeholders.
* You are a code evolving machine, and you will be called 20 times to generate code, and improve the code to achieve the lowest possible validation Wasserstein loss.
* The model defines the possibly stochastic transition function taking the full state, action and predicting the next state, and will be used to fit the observed training dataset.
* You can use any parameters you want however, you have to define these.
* It is preferable to decompose the system into compartments if possible.
* You can use any unary functions, for example log, exp, power etc.
* You can use numpy sampling distributions.
* Under no circumstance can you change the skeleton code function definitions, only fill in the code.
* Make sure your code follows the exact code skeleton specification.
* When defining categorical distributions that are parameterized make it so that the probabilities are automatically normalized as they will be sampled as random values. I.e. normalize the probabilities within the step function.
```

Think step-by-step, and then give the complete full working code. You are generating code for iteration 0 out of 5.

## Supply Chain

You will get a simulator description to code a **`step` function** in NumPy.

System Description:
```
Single-stage Beer Game supply chain environment.

Here you must model the simulation step with the below state and action of

The environment state is represented by a dictionary:
    "inventory": int,              # On-hand units
    "pipeline": list of (int, int), # shipments: (quantity, time_remaining)
    "backlog": int,                # units of unfilled demand
    "t": int                       # current time step

Action: int = is the new order to place with the supplier. That creates a pipeline entry with a lead time sampled from a discrete distribution.

The collected trajectory lasts for 60 time steps (days).
```

Modelling goals:```
* The parameters of the simulator will be optimized to an observed training state-action dataset with the given simulator using simulation-based Inference.
* The observed training dataset has very few samples, and the model must be able to generalize to unseen state-action data.

```
...

Requirement Specification:```
* The code generated should achieve the lowest possible validation Wasserstein loss, of 1e-10 or less.
* The code generated should be interpretable, and fit the dataset as accurately as possible.
```

Skeleton code to fill in:```
class SimulatorStep():
    def __init__(self):
        # TODO: Fill in the code here - define the parameters of the model and make them self here.

    def step(self, state: dict, action: int, rng: np.random.Generator) -> dict:
        # Must include all the logic
        ...
        return next_state

    def get_parameters(self) -> np.ndarray:
        """
        Returns the model parameters as an array.
        """
        # TODO: Fill in the code here

    def set_parameters(self, parameters: np.ndarray):
        """
        Updates the model parameters.

        Args:
            parameters (np.ndarray): Array of parameters to update.
        """
        # TODO: Fill in the code here

    def get_parameters_uniform_prior_min_max -> np.ndarray:
        """
        Returns the uniform prior bounds for the parameters.

        Returns:
            np.ndarray: Array of shape (2, num_parameters) with min and max bounds.
        """
        # TODO: Fill in the code here
```

Useful to know:
```
* The generated code must include the complete `step` function body in NumPy, fully functional, no placeholders.
* You are a code evolving machine, and you will be called 20 times to generate code, and improve the code to achieve the lowest possible validation Wasserstein
    loss.
* The model defines the possibly stochastic transition function taking the full state, action and predicting the next state, and will be used to fit the observed
    training dataset.
* You can use any parameters you want however, you have to define these.
* It is preferable to decompose the system into compartments if possible.
* You can use any unary functions, for example log, exp, power etc.
* You can use numpy sampling distributions.
* Under no circumstance can you change the skeleton code function definitions, only fill in the code.
* Make sure your code follows the exact code skeleton specification.
* When defining categorical distributions that are parameterized make it so that the probabilities are automatically normalized as they will be sampled as random
    values. I.e. normalize the probabilities within the step function.
```

Think step-by-step, and then give the complete full working code. You are generating code for iteration 0 out of 5.
```

# G. Future work and broader impact

**Multiscale or Hybrid Time Steps.**  While we focus on discrete-time or event-based submodules, future expansions could unify different timescales. The LLM can propose which submodules update hourly vs. daily. GFO remains applicable so long as we can run the simulation for each candidate parameter set.

**Complex Domain Constraints.**  If domain constraints are complex (e.g., partial differential equations or advanced PDE-based physics), structural generation can remain feasible but might require specialized symbolic analysis. Alternatively, the user can provide a "template code skeleton" that the LLM must only fill in, ensuring compliance with advanced physics laws.

**Large Submodule Libraries.**  As we scale to hundreds of submodule templates, the search space for $\lambda$ expands exponentially. Methods like "top-$k$ structural proposals" or hierarchical LLM prompting may help prune unpromising structural expansions.

**Diagnostics in Active Learning.**  One could extend the diagnostic function to query new data or domain experts to disambiguate uncertain modules. This aligns with active learning strategies in simulator design.

Overall, the G-Sim approach is flexible and can be adapted for many real-world tasks by appropriately shaping the submodule library, domain constraints, and prompt engineering.

While the discussion in Appendix A and the main text has highlighted immediate applications of G-Sim, there remain abundant opportunities to extend and enrich its capabilities, as well as substantial questions regarding its implications for society and technology. Below, we elaborate on how G-Sim could evolve in the near future, what types of contexts it might be applied to, and which open research problems still need to be addressed to fully realize its potential.

## G.1. Expanding the Scope of Simulation

Although the above use-cases hint at specific challenging scenarios that our framework could address, next we discuss them more "abstractly".

**Combining Multiple Data Sources That Cannot Be Directly Merged.**  One of the powerful, potential properties of G-Sim is its ability to reconcile and unify incomplete, heterogeneous datasets through the non-differentiable parameter-inference and the iterative structural refinement loop. This aligns closely with real-world situations where data is "locked" in separate silos (e.g., due to privacy constraints, mismatched time scales, or proprietary concerns). G-Sim could potentially be applied to a scenario where one dataset tracks daily admissions but omits discharge times, while another captures only average occupancy levels, and a third logs *policy* changes intermittently. Through its flexible calibration framework, G-Sim can synthesize these partial or non-overlapping datasets into a cohesive simulator that is still grounded by real observations, yet remains flexible enough to propose creative structural linkages derived from domain knowledge or large language models (LLMs). Future research could focus on formalizing guarantees on consistency and convergence when combining more than two or three data sources, each with distinct observational modalities or temporal resolutions.

**Continual learning.**  Because the environment is constructed as a composition of submodules, each representing distinct components of the system (e.g., patient arrivals, disease progression, resource management), updates to individual modules can be made without disrupting the entire simulator. This modularity allows the framework to incrementally integrate new data or reflect distributional changes in specific subcomponents (e.g., new patient arrival patterns during a pandemic) while preserving previously learned dynamics in other unaffected modules. Furthermore, the iterative refinement process inherently supports continual adaptation by identifying and isolating discrepancies between the simulator's outputs and newly observed data, prompting targeted adjustments. This ensures that the simulator evolves smoothly to accommodate gradual or abrupt distributional shifts.

**Integrating Data at Multiple Time Scales.**  Real systems often evolve across different granularities. In public health, disease incidence may be reported weekly or monthly, while hospital admissions are tracked daily or even hourly. In supply chains, some processes (e.g., production) update at a weekly scale, while logistics or e-commerce interactions can evolve by the hour or minute. A natural extension would allow G-Sim to handle multi-scale data more explicitly: each module can evolve at its own frequency and pass aggregated signals or constraints to modules on different time scales. In principle, the

modular calibration of G-Sim is well-suited to this task, but further methodological refinement is needed to ensure stable parameter inference, particularly when different submodules have drastically different update cadences.

**Structured Coarse-to-Fine Validation.**    Because G-Sim's calibration can handle sparse or aggregate-level data, there arises the possibility of building a *hierarchy* of simulators that operate at different levels of detail. An initial coarse-grained simulator could match higher-level statistics of a system (e.g., average weekly admissions), and then subsequent layers introduce more detailed or fine-grained modeling (e.g., individual patient characteristics). Iterative refinement would proceed at these nested levels, ensuring each finer simulator remains consistent with coarser macro-level aggregates. This coarse-to-fine validation could also be guided by domain knowledge or by an LLM's feedback on plausible submodule expansions, yielding significantly deeper and more scalable multi-resolution simulators.

## G.2. Potential Uses Beyond Intervention Testing

**Safe Reinforcement Learning and Policy Training.**    Although other works already enable this, G-Sim could serve as a reliable environment to train reinforcement learning (RL) agents, particularly to assess them under distribution shifts and in domains where real-world experimentation is expensive or risky (e.g., healthcare, finance, or transportation). Agents trained in this simulator could be encouraged (through reward shaping or robust policy learning) to avoid certain regions if they might be associated with unacceptable real-world risk.

**Curriculum and Continual Learning.**    Another way to leverage G-Sim is in the spirit of *OMNI-EPIC*-like approaches (Faldor et al., 2024): (1) Start with a simpler environment (fewer modules or constraints, denser rewards) for an agent to learn basic dynamics and decision policies. (2) Gradually make the environment more complex, either by adding submodules discovered by the LLM, by shifting the distribution of parameters or by making the reward sparser or more complex. This could create a curriculum that systematically challenges and develops the agent's capacity for generalization.

## G.3. Limitations and Future Directions

While our approach enables flexible simulator construction even under partial data and complex structural requirements, several open challenges remain.

**Scalability**    First, **scalability** poses a concern: gradient-free searches can become computationally expensive as the number of parameters or submodules grows, especially for large or high-resolution systems. Parallelization partially addresses this but may still be infeasible for multi-scale, multi-region simulators with thousands of discrete or stochastic elements. More efficient search techniques and tailored surrogate models—potentially leveraging hierarchical or active-learning schemes—offer promising ways to accommodate such complexity. Future work could also explore exploiting the causal structure hypothesized by the LLM to train sequentially the existing sub-components that might be independent of others conditioned on certain features to alleviate complexity.

**Structural Coverage by the LLM**    Second, **structural coverage by the LLM** is not guaranteed when domain knowledge is sparse or highly specialized. If the language model never proposes a submodule or coupling essential for fidelity, calibration cannot recover the correct dynamics. Domain experts must still review and guide the simulator-building process, particularly in safety-critical applications like healthcare or critical infrastructure. Iterative prompt design (Sahoo et al., 2024), retrieval-augmented generation (i.e., letting the LLM consult curated references) (Lewis et al., 2021), and manual injection of known constraints may further improve coverage.

**Multi-Timescale and Partial Observability**    Third, multi-timescale and partial observability introduce subtleties in scoring and inference. While the method theoretically should already supports heterogeneity in data type and coverage, future work could refine diagnostic criteria for simulator mismatch (e.g., across daily, weekly, and monthly scales) or more seamlessly piece together observations spanning asynchronous submodules.

**Biases from Data and LLM Priors**    Fourth, biases from both data and LLM priors warrant careful scrutiny. When historical data reflect inequitable policies or an LLM's training corpus omits certain domain factors, the resulting simulator may embed inaccurate or unfair assumptions (Wei et al., 2025). Strengthening mechanisms that detect such issues—perhaps by incorporating fairness constraints or domain-specific plausibility checks—will be essential before deploying these simulators in sensitive real-world settings.

## G.4. Ethical Considerations and Mitigation Strategies

The deployment of simulators like G-Sim, especially in sensitive domains such as healthcare, logistics, and epidemic planning, carries significant ethical responsibilities. The potential for these simulators to inform high-stakes decisions requires a proactive approach to identifying and mitigating risks, particularly those related to bias and societal impact.

**Potential Risks and Biases.** The primary risks arise from potential biases embedded within either the foundational LLMs used for structural generation or the empirical data used for calibration. LLMs may carry biases from their vast training corpora, potentially leading to simulator structures that overlook or misrepresent certain sub-populations or dynamics. Similarly, historical datasets can reflect past inequities or contain measurement errors, which, if unaddressed during calibration, could lead to simulators that perpetuate or even amplify these biases, resulting in unfair or harmful *policy* recommendations.

**Mitigation Strategies in G-Sim.** Our framework is designed with several features to mitigate these risks and promote responsible use:

- **Transparency and Inspectability:** G-Sim generates simulator structures as explicit, human-readable code. This transparency allows domain experts to thoroughly inspect, understand, and verify the underlying mechanisms and assumptions, facilitating accountability and trust.
- **Expert-in-the-Loop:** We strongly emphasize the indispensable role of domain experts. G-Sim is designed to augment, not replace, human expertise. Experts are crucial for providing context, validating LLM proposals, interpreting diagnostic results, stress-testing the simulator against known edge cases, and ensuring its outputs align with established knowledge and ethical guidelines.
- **Iterative Diagnostics and Refinement:** The framework's diagnostic loop enables rigorous evaluation against empirical data and domain-specific constraints. This process can help identify and rectify biases or inaccuracies. Future extensions could explicitly incorporate fairness metrics or adversarial checks into these diagnostics.
- **Uncertainty Quantification:** Integrating methods like SBI allows for approximate (albeit with challenges already discussed) quantification of uncertainty in parameter estimates. Communicating this uncertainty is vital for decision-makers to understand the confidence levels associated with simulation outcomes.

**Responsible Deployment.** G-Sim should be viewed as a powerful tool for exploration and decision support, not an infallible oracle. Its deployment, particularly in real-world applications with societal consequences, must be accompanied by rigorous validation, continuous oversight, and a transparent articulation of its assumptions and limitations. Ensuring that these simulators support ethical, equitable, and beneficial outcomes remains a shared responsibility between developers, domain experts, and decision-makers.

# H. Evaluation Metrics

To rigorously assess how closely the learned simulators align with the ground-truth generative process, we employ evaluation metrics designed to compare probability distributions. Our primary metrics are the **Wasserstein distance** (Kantorovich, 1960) (also known as Earth Mover's Distance) and the **Maximum Mean Discrepancy (MMD)** (Gretton et al., 2012). These metrics are particularly well-suited for our benchmark tasks, many of which exhibit stochastic or discrete transitions where capturing the full distributional shape is crucial.

Concretely, for each initial state $\mathbf{x}_0$ in our test set $\mathcal{D}_{\text{test}}$ (along with any corresponding actions or controls), we generate two sets of next-state samples:

1. **Ground-Truth Samples:** We run the ground-truth simulator $N$ times (typically $N = 1000$) from $\mathbf{x}_0$ to obtain $\{\mathbf{x}_{1,i}^{(g)}\}_{i=1}^N$.

2. **Comparison Samples:** We run the candidate simulator $N$ times from the same $\mathbf{x}_0$ to get $\{\mathbf{x}_{1,i}^{(c)}\}_{i=1}^N$.

We then compute the Wasserstein distance and MMD between these two empirical distributions, $\{\mathbf{x}_{1,i}^{(g)}\}$ and $\{\mathbf{x}_{1,i}^{(c)}\}$. We repeat this across all initial states in $\mathcal{D}_{\text{test}}$ and report the average distances.

The **Wasserstein distance** intuitively captures the minimum "cost" (or work) required to transform one distribution into the other, providing a holistic comparison of their shapes. **MMD** measures the distance between distributions by comparing their mean embeddings in a Reproducing Kernel Hilbert Space (RKHS); a zero MMD implies the distributions are identical. Both offer significant advantages over simpler pointwise metrics.

**Limitations of Mean Squared Error (MSE).** MSE measures the average squared difference between individual predictions and ground-truth values. When dealing with stochastic transitions, a simulator aiming to minimize MSE will be incentivized to predict the *mean* of the next-state distribution. This approach fails to capture the inherent variability and potentially multi-modal nature of the true process. A simulator that perfectly predicts the mean but ignores the variance or shape of the distribution would achieve a low MSE but would be a poor representation of the system's dynamics. Therefore, we prioritize Wasserstein and MMD for assessing the *distributional fidelity* of our generated simulators.

We run each method for ten independent seeds (unless stated otherwise) and report the mean and 95% confidence intervals for each metric. All experiments and training were performed using a single Intel Core i9-12900K CPU @ 3.20GHz, 64GB RAM with an Nvidia RTX3090 GPU 24GB.

# I. Additional Experiments

To further validate the capabilities and robustness of G-Sim, we conducted several additional experiments. These investigations focused on (1) out-of-distribution generalization in the supply chain environment, (2) the structural accuracy of the generated simulators using causal discovery metrics, (3) a comparison between Gradient-Free Optimization (GFO) and Simulation-Based Inference (SBI) for parameter calibration, (4) the computational performance of G-Sim relative to baselines, and (5) an illustration of the iterative refinement process.

## I.1. Out-of-Distribution Generalization: Supply Chain Backlog

A critical requirement for effective simulators is their ability to generalize to scenarios not seen during training, particularly those involving interventions or shifts in system dynamics. We tested G-Sim's out-of-distribution (OOD) performance by varying the lead times $\ell$ in the supply-chain environment beyond the range observed during the initial training phase.

Figure 4 displays the backlog trajectories for both the ground-truth environment and the G-Sim-generated simulator under increasing lead times. As observed:

- For short lead times ($\ell = 1, 2, 3$), consistent with or close to the training data, both the true system and G-Sim maintain a near-zero backlog, indicating accurate calibration in the known regime.

- As the lead time increases ($\ell = 4, 5, 6$), creating a significant OOD challenge, both systems exhibit pronounced backlog spikes due to delayed inventory replenishment.

- Crucially, G-Sim plausibly predicts both the timing and magnitude of these spikes, even though it was not explicitly trained on these longer delays.

This strong OOD performance underscores G-Sim's ability to capture the underlying causal mechanisms of the supply chain. By leveraging the LLM to propose a structurally plausible model (including inventory, pipeline, and backlog dynamics) and then calibrating its parameters, G-Sim learns a representation that is not only correlational but also reflects the system's core dynamics. This structural grounding enables it to extrapolate reliably, a capability often lacking in purely data-driven models and essential for evaluating "what if" scenarios involving *policy* shifts or external shocks, such as supply disruptions.

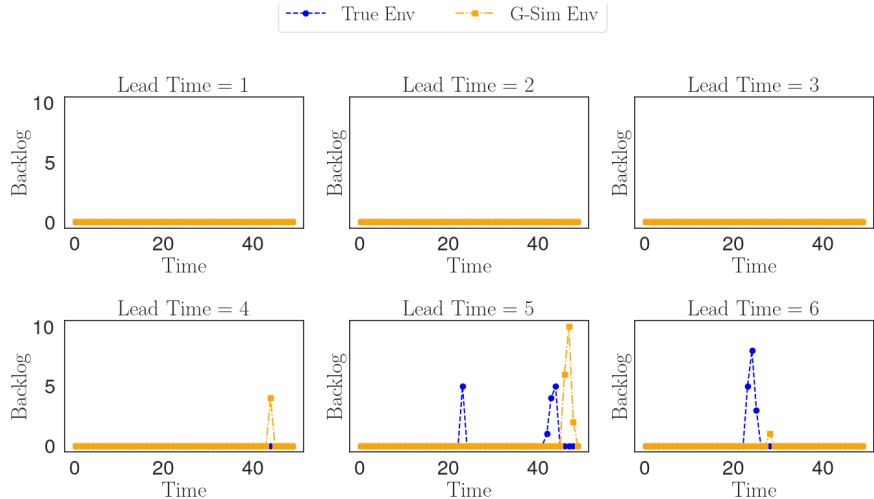

Figure 4. Backlog vs. time in the supply-chain environment for varying lead times $\ell$. Blue circles and dashed lines indicate the true environment's backlog, while orange squares and dash-dot lines show the G-Sim environment. For short lead times ($\ell = 1, 2$), both maintain near-zero backlog. Longer lead times ($\ell = 4, 5, 6$) cause backlog spikes at similar times, indicating G-Sim closely tracks the real system's OOD dynamics.

*Table 4.* Causal Discovery Metrics for G-Sim across three environments. We report Structural Hamming Distance (SHD) ↓ and F1 Score (%) ↑, averaged over five random seeds, with ± denoting 95% confidence intervals. G-Sim demonstrates high structural accuracy, recovering near-perfect or perfect causal graphs.

| Metric | COVID-19 Value | Supply Chain Value | Hospital Beds Scheduling Value |
|---|---|---|---|
| F1 Score (%) ↑ | $95.5 \pm 4.09$ | $93.3 \pm 6$ | $100.0 \pm 0.0$ |
| SHD ↓ | $1.5 \pm 1.35$ | $0.333 \pm 0.3$ | $0 \pm 0$ |

## I.2. Structural Accuracy via Causal Discovery Metrics

A core claim of G-Sim is its ability to generate simulators with causally sound structures. To quantify this, we evaluated the structural accuracy of the simulators generated by G-Sim against the known ground-truth causal graphs for each environment. We used two standard metrics from causal discovery:

- **Structural Hamming Distance (SHD)**: Measures the number of edge insertions, deletions, or reversals needed to transform the predicted graph into the true graph. Lower values are better.

- **F1 Score**: The harmonic mean of precision and recall for edge prediction, providing a balanced measure of edge discovery accuracy. Higher values (up to 100%) are better.

Table 4 presents these metrics for G-Sim across the three benchmark environments. The results demonstrate that G-Sim achieves high structural accuracy:

- For the **Hospital Bed Scheduling** environment, G-Sim achieves a perfect F1 score (100%) and zero SHD, indicating an exact match with the ground-truth causal structure.

- For **COVID-19** and **Supply Chain**, G-Sim achieves F1 scores above 90% and very low SHD values. This indicates that G-Sim reliably captures the essential causal relationships, even in environments with stochasticity and partial observability.

These quantitative results support our claim that G-Sim, by combining LLM-driven structural reasoning with empirical calibration, can reliably estimate the underlying causal model, a key factor in its strong OOD generalization and its suitability for *policy* intervention studies.

## I.3. Parameter Calibration: GFO vs. Simulation-Based Inference (SBI)

We integrated Simulation-Based Inference (SBI) into G-Sim as an alternative to GFO for parameter calibration. SBI methods, like Neural Posterior Estimation (NPE), aim to learn a full posterior distribution over simulator parameters, $p(\omega|\mathcal{D})$, rather than just a point estimate. This offers principled uncertainty quantification, which is crucial given that the LLM might propose imperfect structures.

We compared G-Sim – ES with G-Sim – SBI on the COVID-19 SIR task, using the same LLM-generated model structure for both. Table 5 summarizes the results.

*Table 5.* Comparison of G-Sim – ES and G-Sim – SBI on the COVID-19 SIR environment. We report computation time and test set performance (MSE, MMD, Wass. distance). Lower values are better. Results are averaged over five seeds ± 95% CI.

| Metric | G-Sim – ES | G-Sim – SBI |
|---|---|---|
| Computation Time (s) ↓ | $55.7 \pm 0.399$ | $63.9 \pm 2.49$ |
| Test MSE ↓ | $3.11 \pm 0.0558$ | $1.11\text{e+}03 \pm 491$ |
| Test MMD ↓ | $0.0583 \pm 0.0197$ | $0.07 \pm 0.0245$ |
| Test WASS ↓ | $0.814 \pm 0.0346$ | $1.29 \pm 0.101$ |

Key observations include:

*Table 6.* Training time (seconds) and number of trainable parameters for all benchmark methods.

| Method | Train Time (s) | # Parameters |
|---|---|---|
| DyNODE | 328 | 34,951 |
| SINDy | 3 | 175 |
| RNN | 33 | 572,507 |
| Transformer | 95 | 2,560,353 |
| GeneticProgram | 134 | 26 |
| G-Sim-ES Abl. ZeroShot | 366 | 4 |
| G-Sim-ES Abl. ZeroShotOptim | 426 | 4 |
| **G-Sim – ES** | 2,314 | 16 |

- **Computational Cost**: SBI has a slightly higher computational time, but both methods are comparable, operating within similar simulation budgets.

- **Predictive Accuracy**: In these initial runs, G-Sim – ES achieved lower (better) MSE and Wasserstein distances. This may reflect the direct optimization objective of ES versus the inference objective of SBI, or potential sensitivities in the SBI implementation to model misspecification.

- **Uncertainty Quantification**: The primary advantage of SBI is its ability to provide posterior distributions, a crucial feature that GFO lacks. However, as discussed in Appendix B.4, interpreting these posteriors requires care when the model structure is not guaranteed to be correct.

While G-Sim – ES shows an edge in point-estimate accuracy in this setup, the integration of SBI significantly strengthens G-Sim by enabling uncertainty quantification, albeit with important caveats.

### I.4. Computational Performance: Training Times and Parameters

To provide context on the computational demands of G-Sim compared to standard baselines, we measured training times and the number of trainable parameters for each method (Table 6).

G-Sim has the longest training time. This is expected as it involves multiple iterations of LLM calls, code generation, simulation runs, and calibration, rather than a single pass of gradient descent. However, G-Sim often results in simulators with significantly fewer intrinsic parameters (e.g., 16 in the final model) compared to large neural network models like RNNs or Transformers. This reflects G-Sim's focus on discovering interpretable, parsimonious structures, where complexity lies in the generative search process rather than in a massive parameter count. SINDy and Genetic Programs are fast and have few parameters, but showed weaker predictive performance in our main results.

### I.5. Iterative Refinement by G-Sim: An Example

The core of G-Sim's innovation lies in its iterative refinement loop, where diagnostic feedback guides the LLM to improve the simulator's structure. To illustrate this, we provide a segment of the G-Sim logs from the COVID-19 task. These logs show the LLM's reflections and the code changes made across iterations based on performance metrics and feedback. We observe how the LLM proposes modifications, such as adding overdispersion or adjusting functional forms, to better match the observed data.

```
...
Val Wasserstein Loss: 0.793 (Where the val MSE loss per dimension is susceptible val MSE loss: 2.85, infected val MSE loss: 4.7, recovered val MSE loss: 2.04)
        Iteration: 1
###
...
import numpy as np

class SimulatorStep():
        def __init__(self):
                # alpha: offset, beta: slope, gamma: recovery, k_disp: overdispersion
                self.alpha, self.beta, self.gamma, self.k_disp = 0.0, 0.01, 0.1, 1.0
...
...
optimized_parameters = [-3.54, 4.47, 0.073, 24.45]
###
...
Val Wasserstein Loss: 0.404 (Where the val MSE loss per dimension is susceptible val MSE loss: 1.89, infected val MSE loss: 3.95, recovered val MSE loss: 2.03)
        Iteration: 0
```

```
###
```
import numpy as np

class SimulatorStep():
        def __init__(self):
                self.beta, self.gamma = 0.01, 0.1
...
```
optimized_parameters = [0.457, 0.079]
###

Please reflect on how you can improve the code...
[Assistant] 1) Introduce an overdispersion (negative binomial or similar) for both infections and recoveries, not just for infections. This helps capture
        variability in how individuals recover and can reduce error in the infected and recovered compartments.
2) Add an offset plus slope for the recovery process (for example, a logistic function or a piecewise linear function in I/N) rather than using a fixed gamma.
...
```

This iterative process, documented in the logs, allows G-Sim to move beyond an initial, potentially flawed, structural proposal towards a simulator that is both structurally plausible and empirically aligned.

