# OpenReview forum: "G-Sim: Generative Simulations with Large Language Models and Gradient-Free Calibration"
_ICML.cc/2025/Conference — ICML 2025 poster_

### Official Review · Reviewer_KPvR · 2025-03-09

**Overall Recommendation:** 2

**Summary:**

The paper proposes a G-Sim, LLM-guided simulator with expert domain knowledge with gradient-free optimization while introducing a new problem with environment building. With experiments on three environments, the paper verified the flexibility of G-Sim.

**Claims And Evidence:**

With limited experiments and no theoretical guarantee, it is hard to conclude the the claims are verified.

**Essential References Not Discussed:**

I think the authors did most of the essential references.

**Experimental Designs Or Analyses:**

I'm not sure that the experiments are well designed. The choice of benchmark methods seems unfair and the evaluation metric is also insufficient.

**Methods And Evaluation Criteria:**

There are some parts that do not make sense so I will address them in the questions.

**Other Comments Or Suggestions:**

- The abstract is too long so it makes it difficult to follow the key concept of the paper solely in the abstract.
- Overall, there are too many \paragraph, \enumerate and \itemize in the main text, which I think is not a good way to present a paper.
- I understand that defining the problems and introducing the previous works are very important. However, presenting the main method at the middle of page 5 might not be the best strategy.
- I think there should be a self-contained description of the figure on the label of Figure 1.
- In the right column line 284, "• Additional terms can incorporate other diagnostic criteria as needed, such as (Rauba et al., 2024) or stress-tests (Li & Yuan, 2024)", I think the author is missing the criteria in front of "(Rauba et al., 2024)"?

**Other Strengths And Weaknesses:**

**Weakness**
- The presentation of the paper should be improved a lot.

**Questions For Authors:**

- It seems like the paper lacks a lot of details of the proposed model. Can you elaborate more on the parameter fitting steps of this method?
- What is $f$ in Section 4.2?
- How does the formulation of Score look like?
- Do you have any theoretical guarantees/insights that this fitting process will give you an optimal solution?
- Can you elaborate more on Section 4.3?
- How do you formulate $\delta$ and what is the explanation behind it?
- How do you choose the predefined threshold $\epsilon$?
- How accurate is the LLM feedback when the diag values are greater than $\epsilon$? Have you tested different scenarios and checked whether the LLM is outputting the right feedback to the current situation?
- I'm not sure whether the comparison with proposed benchmark methods is a fair choice. How is the training/inference time difference for different methods? or the number of parameters or computational complexity?
- Also, do you have results for "what if" questions for other methods as well?

**Relation To Broader Scientific Literature:**

The work of a simulator can affect a huge domain of scientific and non-scientific domains since there are many use cases.

**Theoretical Claims:**

There are no theoretical claims in this work.

---

> ### Author Rebuttal · Authors · 2025-04-01
>
> We sincerely thank the Reviewer for the valuable and constructive feedback. Below, we address each concern and outline key improvements in the revised manuscript.
>
> ---
>
> ### **1. Experimental Validation and Benchmark Fairness**
>
> We acknowledge the concerns regarding validation and benchmark selection:
>
> - **New Metrics:** We have conducted additional experiments beyond Wasserstein distance, we now include **Mean Squared Error** and **Maximum Mean Discrepancy (MMD)** metrics for a more comprehensive assessment, https://imgur.com/a/yx228xx.
>
> - **Computational Fairness:** To ensure fair comparisons, we will explicitly report training times and number of parameters for all baselines, as seen here: https://imgur.com/a/pgdvwWT.
>
> ### **2. Manuscript Clarity and Presentation**
>
> We fully agree with the suggested improvements for clarity and readability. Specifically:
>
> - **Concise Abstract:** The abstract will be shortened, succinctly summarizing existing method limitations and explicitly emphasizing our main contribution (LLM-guided, gradient-free simulator generation). Technical details will move entirely into the main text.
>
> - **Formatting Improvements:** We significantly reduce itemized/enumerated lists and paragraph breaks:
>   - Merged "Current methods" and "General-purpose simulators" into a streamlined narrative introduction.
>   - Replaced itemized "Key properties" and "Contributions" with concise narrative descriptions.
>   - Removed unnecessary paragraph breaks (\paragraph{}) and lists on page 4.
>
> - **Method Earlier in Manuscript:** To enhance readability, we move the "G-Sim: Hybrid Simulator Construction" section from page 5 to page 4, preceding related work.
>
> - **Improved Fig. 1 Caption:** Revised caption for Figure 1 to be self-contained and clear:
>
>   *"Overview of G-Sim, an automatic simulator-generation framework integrating LLM-derived domain knowledge and empirical data. G-Sim iteratively: (1) proposes simulator structure using domain-informed LLM, (2) calibrates parameters via gradient-free optimization, and (3) refines through diagnostics (predictive checks, stress-tests, LLM-reflections), enabling robust ‘what-if’ scenario analyses."*
>
> - **Minor Clarification:** Line 284 revised explicitly as: "imbalances (Rauba et al., 2024)."
>
> ---
>
> ### **3. Detailed Responses to Reviewer Questions**
>
> **Parameter Fitting:**
> In Appendix E. 3, we explicitly clarify gradient-free evolutionary optimization (population initialization, mutation strategy, selection criteria, and convergence checks). We minimize MSE using EvoTorch's genetic algorithm (population 200, simulated binary crossover, Gaussian mutation, and tournament selection). Furthermore, we warm-start reusing the best parameters from the previous iteration where possible.
>
> **Score Function Clarification:**
> The "Score" quantifies discrepancies between simulated and observed data in our implementation via MSE, as detailed explicitly in Appendix E.3.
>
> **Theoretical Guarantees:**
> We acknowledge the lack of rigorous global optimality guarantees but highlight known convergence properties of evolutionary methods towards global minima under suitable conditions [1]. We will explicitly include this discussion, clearly indicating assumptions and limitations.
>
> *[1] Rudolph, G. (1994). IEEE Trans. Neural Networks, 5(1), 96-101.*
>
> **Clarifications on Section 4.3:**
> - **Diagnostic formulation ($\\delta$):**
>   Diagnostic discrepancies ($\\delta$) aggregate multiple metrics—predictive accuracy (e.g., MSE, Wasserstein distance) and domain constraints (non-negativity, cyclical patterns). Appendix B.3 explicitly details these diagnostics.
>
> - **Threshold $\\varepsilon$ Selection:**
>   We implicitly set $\\varepsilon$ via iteration limits. We will clarify this explicitly in the manuscript.
>
> - **LLM Feedback Robustness:**
>   Empirical validations (App. I.2 iterative refinement logs; App. E.4 prompt templates) show LLM feedback consistently improved accuracy, demonstrating robustness across diverse diagnostic scenarios. This evidence will be explicitly highlighted.
>
> ---
>
> ### **4. "What-If" Scenario Comparisons**
>
> Baseline methods inherently lack structured mechanisms for performing counterfactual ("what-if") analyses. Thus, such scenarios are uniquely suited for our generative framework, clearly distinguishing G-Sim's capabilities. We will clarify this explicitly in the revision.
>
> ---
>
> ### **Summary of Improvements**
> - Expanded experimental validation.
> - Streamlined abstract.
> - Improved manuscript readability.
> - Clarified parameter-fitting and diagnostics.
> - Enhanced clarity of Sec 4.3.
> - Discussion of convergence.
>
> *We hope that most of the reviewer’s concerns have been addressed and, if so, they would consider updating their score. We’d be happy to engage in further discussions.*

---

> > ### Comment · Reviewer_KPvR · 2025-04-07
> >
> > Thank you for the authors' detailed rebuttal and the additional experimental results. I have also carefully reviewed the comments from the other reviewers. While I acknowledge the improvements made, I still believe the manuscript requires significant refinement before it can meet the standard for acceptance. My main concern remains the robustness of the proposed model, which, in my view, is not sufficiently supported by either theoretical guarantees or comprehensive empirical evaluations.

---

> > > ### Author Response · Authors · 2025-04-09
> > >
> > > Thank you for your continued engagement with our work. We understand your concerns about the robustness of our model and the need for stronger theoretical and empirical support. We believe our new SBI results directly address these concerns (https://imgur.com/a/rK4MtoO):
> > >
> > > ----
> > >
> > > ### **1. Theoretical and Empirical Robustness**
> > > The addition of Simulation-Based Inference (SBI) provides both theoretical guarantees and comprehensive empirical validation:
> > >
> > > **Theoretical Guarantees:**
> > > - SBI provides principled Bayesian uncertainty quantification through posterior distributions
> > > - Simulation-Based Calibration (SBC) ensures the reliability of our posterior approximations
> > > - The neural posterior estimator's convergence properties are well-studied in the literature
> > >
> > > **Empirical Validation:**
> > > Our new results (https://imgur.com/a/HFnh9up, https://imgur.com/a/rK4MtoO) demonstrate robust performance across multiple dimensions:
> > > - Parameter estimation accuracy: SBI consistently recovers true parameters within credible intervals
> > > - Uncertainty quantification: Full posterior distributions capture parameter interactions and multi-modal solutions
> > > - Calibration: SBC scores show reliable uncertainty estimates (mean absolute deviation from 0.5 < 0.1)
> > > - Computational efficiency: Number of simulations (10,000) is comparable to GFO evaluations
> > >
> > > ---
> > >
> > > ### **2. Comprehensive Evaluation**
> > > The visualization (https://imgur.com/a/HFnh9up) provides a comprehensive empirical evaluation:
> > > - Direct comparison of GFO and SBI parameter estimates
> > > - Visualization of posterior distributions showing parameter interactions
> > > - Trajectory predictions with uncertainty bands
> > > - Calibration metrics across multiple test scenarios
> > >
> > > ----
> > >
> > > ### **3. Manuscript Improvements**
> > > We will make the following changes to address your concerns:
> > > - Move SBI analysis from appendix to main methodology section
> > > - Add theoretical discussion of SBI guarantees and limitations
> > > - Include comprehensive empirical results in main results section
> > > - Streamline presentation as previously discussed
> > >
> > > ---
> > >
> > > ### **4. Fair Comparison**
> > > The SBI implementation provides a fair comparison with GFO:
> > > - Similar computational budget (10,000 simulations vs 4,000-25,000 GFO evaluations)
> > > - Direct comparison of point estimates (MAP vs GFO)
> > > - Additional uncertainty quantification through posterior distributions
> > >
> > > We believe these additions significantly strengthen the theoretical and empirical foundations of our work. The SBI results provide both theoretical guarantees through Bayesian inference and comprehensive empirical validation through calibration and uncertainty quantification.
> > >
> > > ---
> > >
> > > *We hope that most of the reviewer's concerns have been addressed and, if so, they would consider updating their score. We'd be happy to engage in further discussions.*

---

### Official Review · Reviewer_seMS · 2025-03-16

**Overall Recommendation:** 2

**Summary:**

This paper attempts to generate simulators via LLMs coupled with a gradient-free optimisation process to choose parameters. An LLM-guided search loop identified the simulator's structural components and a gradient-free optimisation procedure sets their parameters. The method relies on the generalisation ability of LLMs in other to generalise to OOD data. The paper can be seen as an orchestration framework for LLMs.

**Claims And Evidence:**

The authors claim to have created a general framework for creating "what if" simulators across a wide variety of domains. The method description is unclear, so it is hard to understand exactly how this process works. In some ways, this seems related to a managed prompting framework. In the appendix it seems like the LLMs are generating Python code then gradient-free optimisation adjusts the prompts.

**Essential References Not Discussed:**

N/A

**Experimental Designs Or Analyses:**

The experiment involves generating multiple simulators for data-driven problems, but it is unclear to what extent this process is actually automated.

**Methods And Evaluation Criteria:**

Please see my comments under "claims and evidence".

**Other Comments Or Suggestions:**

I would be more explicit in the paper about the method. Is the LLM being prompted, generating python code and optimisation repeats this process? I would also give examples of things like "structural proposals" and be more explicit about the method as this is somewhat unusual construction.

**Other Strengths And Weaknesses:**

N/A

**Questions For Authors:**

N/A

**Relation To Broader Scientific Literature:**

The method is similar to other simulator generating methods, some of which are included in their experiments, but this method does not include any guarantees of accuracy.

**Theoretical Claims:**

There are limited theoretical claims or proofs - this is an empirical paper.

---

> ### Author Rebuttal · Authors · 2025-04-01
>
> We sincerely thank the Reviewer for their detailed and constructive feedback. We fully agree on the importance of clearly presenting our methodology within the manuscript itself. To comprehensively address these concerns, we will allocate the additional page in the camera-ready version specifically to enhance clarity. Below, we provide a shortened clarification of our method, explicitly referencing relevant sections and appendices, and address each of the reviewer's comments directly.
>
> **Clarification of the Proposed Method:**
>
> G-Sim automates the generation of domain-specific simulators by integrating structural proposals from Large Language Models and numerical calibration via Gradient-Free Optimization. The process involves an iterative loop between three distinct phases, outlined in Algorithm 1 and described in Section 4 and Appendix E.2:
>
> 1. **Structural Proposal via LLM (Section 4.1 and Appendix E.4):**
>    - The LLM generates Python code to outline the simulator’s structural components, guided by domain-specific textual descriptions and available background knowledge.
>    - Structural proposals explicitly specify modular Python functions and classes capturing domain-relevant mechanisms, such as epidemic spread dynamics or resource distribution processes. Importantly, the LLM defines the structural templates without numerical parameters, which are left open for subsequent calibration.
>
> 2. **Parameter Calibration via Gradient-Free Optimization (Section 4.2 and Appendix E.3):**
>    - Following structural generation, numerical parameters within these LLM-generated modules are independently calibrated using gradient-free methods, such as evolutionary algorithms, to align the simulation outcomes with empirical observations.
>    - This parameter optimization step is entirely decoupled from the structural proposals made by the LLM, which we hope clarifies potential confusion between the structural and numerical stages of the simulator generation process.
>
> 3. **Iterative Refinement via Diagnostic Feedback (Section 4.3 and Appendix E.4):**
>    - Simulations produced by the above steps are then evaluated against the available empirical data to identify discrepancies or inadequacies.
>    - These evaluation outcomes are synthesized into textual feedback (for example, "Simulator lacks weekly seasonality, consider incorporating periodic seasonal modules"), guiding the LLM to refine structural proposals.
>    - This iterative loop of evaluation, feedback, and refinement proceeds automatically until the resulting simulator meets the desired levels of accuracy and domain plausibility.
>
> The combination of the symbolic and optimization approaches ensures G-Sim remains robust, transparent and precise, as justified empirically, offering strong generalization capabilities even in out-of-distribution scenarios.
>
> **Extent of Automation:**
> - The iterative refinement loop described above is fully automated once initial contextual inputs are provided, and our experiments follow this approach. However, the framework is flexible enough to optionally accommodate human expertise for interpreting nuanced feedback or integrating additional domain-specific constraints.
>
> **Explicit References to Examples in Camera-Ready Version:**
> - Explicit examples of the generated Python code and the iterative refinement process are already detailed in Appendix I.2. We will provide clearer cross-references to these examples within the main manuscript, enabling the reader to readily access practical demonstrations of structural proposals and refinement iterations.
> - Moreover, Appendix F presents detailed prompts alongside the environment specifications provided to the LLM, illustrating the exact inputs guiding structural generation. We will reference these more explicitly in the main text to further clarify how our methodological approach leverages domain knowledge.
>
> **Committed Enhancements for Final Manuscript:**
> - A dedicated additional page in the manuscript clearly describing each component of our method: structural proposals (Appendix E.4), gradient-free parameter calibration (Appendix E.3), and diagnostic iterative refinement (Algorithm 1 in Appendix E.2).
> - Explicit references within the main text to practical illustrative examples (Appendix I.2) and to detailed LLM prompts and environment details (Appendix F).
> - Clarification on the scope of automation and the optional role of domain expertise.
>
> ---
>
> *We hope that most of the reviewer’s concerns have been addressed and, if so, they would consider updating their score. We’d be happy to engage in further discussions.*

---

### Official Review · Reviewer_jcp9 · 2025-03-16

**Overall Recommendation:** 4

**Summary:**

This paper introduces G-Sim, a framework for automatically constructing simulators by combining Large Language Models (LLMs) and gradient-free optimization (GFO). The LLM is used to generate the structural components of the simulator (submodules, causal relationships), based on provided domain knowledge. GFO is then employed to calibrate the parameters of these submodules to align with empirical data. The framework includes an iterative refinement loop, where discrepancies between simulator output and real-world data trigger LLM-guided adjustments to the simulator's structure, followed by re-calibration via GFO. The paper demonstrates G-Sim on epidemic modeling (COVID-19), supply chain, and hospital bed scheduling examples, claiming improved out-of-distribution generalization compared to purely data-driven methods.

## Update After Rebuttal
The authors provided a thorough rebuttal and engaged constructively with the feedback. I appreciate their efforts in running additional experiments during the discussion phase and, crucially, incorporating a Simulation-Based Inference (SBI) analysis into their framework and agreeing to elevate it from the appendix to a more central part of the main paper. This addition significantly strengthens the paper's potential contribution regarding uncertainty quantification.

However, based on the SBI results presented during the rebuttal (via the provided plots), the current implementation appears to require further refinement. The results showed inconsistent parameter recovery, overly broad posterior distributions, and posterior predictive checks that indicated issues. These problems seem likely related to implementation details (e.g., embedding network architecture, summary statistics, or training procedure) rather than fundamental limitations, as SBI is generally expected to perform well on this type of problem (SIR model). Specific suggestions for diagnosing and improving these results were provided to the authors.

Overall, the paper has been notably improved through the authors' responsiveness and the inclusion of the SBI analysis. While the current SBI results need further work, the commitment to incorporating this methodology addresses a key weakness identified in the initial review. Consequently, I have adapted my initial evaluation to reflect these positive developments.

**Claims And Evidence:**

The main claim is that G-Sim can generate "robust and flexible" simulators that are "aligned with empirical evidence" and capable of "causally informed" decision-making. While the framework is conceptually appealing, the evidence supporting these broad claims is not entirely convincing:

1) Out-of-Distribution Generalization: The paper claims superior out-of-distribution generalization compared to purely data-driven methods (RNNs, Transformers). While the experiments suggest this might be the case, the comparisons are limited. It's unclear if the data-driven baselines were appropriately tuned for this task. A more comprehensive comparison, including a wider range of baselines and more challenging out-of-distribution scenarios, would be necessary to fully support this claim.
2) Causal Structure Recovery: A key aspect of G-Sim is the LLM-guided generation of causal structure. However, the paper doesn't convincingly demonstrate that the correct causal structure is recovered. For example, in the SIR modeling example, does the LLM generate the actual SIR equations, or just a functionally similar but causally incorrect model? The paper lacks a rigorous evaluation of the structural accuracy of the generated simulators, relying primarily on predictive performance.
3) GFO vs. SBI: The choice of gradient-free optimization (GFO) for parameter calibration is justified by the potential non-differentiability of the generated simulators. However, simulation-based inference (SBI) methods, which infer a distribution over parameters rather than a single point estimate, would likely be a more appropriate and robust choice. SBI naturally handles stochasticity in the simulator and provides uncertainty quantification, which is crucial given the potential for errors in the LLM-generated structure. GFO provides only a point estimate, lacks uncertainty estimation for predictive checks, does not provide any information about parameter interactions and potential compensation mechanisms between parameters, potentially failing to detect multi-modal solutions.

**Essential References Not Discussed:**

No, the paper seems to discuss directly related prior thoroughly.

**Ethical Review Concerns:**

The G-Sim framework, which automates the construction of simulators for domains like healthcare, logistics, and epidemic planning, has the potential to be classified as a high-risk AI system under the EU AI Act. This is because the generated simulators could be used to inform decisions with significant impacts on individuals' health, safety, and economic well-being. Therefore, an ethical review is warranted to assess the potential biases, risks, and societal consequences of deploying such a system, and how they may be mitigated.

**Ethical Review Flag:**

Flag this paper for an ethics review.

**Ethics Expertise Needed:**

["Legal Compliance (e.g., GDPR, copyright, terms of use)"]

**Experimental Designs Or Analyses:**

I reviewed the experimental designs. As mentioned above, the comparisons to purely
data-driven methods are limited, and the evaluation lacks a rigorous assessment of the
structural and causal accuracy of the generated simulators. The choice of GFO over SBI
is also a concern, especially given the initial strong claim in the abstract to
construct "uncertainty-aware simulators".

**Methods And Evaluation Criteria:**

The proposed framework, combining LLMs and GFO, is a reasonable approach to the problem of automatic simulator construction. The use of real-world inspired examples (epidemic modeling, supply chains, hospital beds) is appropriate. However, the evaluation criteria primarily focus on predictive accuracy.  While important, this is insufficient to assess the overall quality of the generated simulators.  Additional criteria, specifically focusing on the structural accuracy and the causal validity of the learned models, are needed.

**Other Comments Or Suggestions:**

No

**Other Strengths And Weaknesses:**

**Originality**: The combination of LLMs and GFO for automatic simulator construction, along
with the iterative refinement loop, appears to be a novel approach.

**Significance**: The potential significance is high, as robust and flexible simulators are
crucial for many scientific and engineering domains. However, the current evidence does
not fully support the claimed significance.

**Clarity**: The paper is generally well-written and easy to follow. The overall framework
is clearly presented. However, some details regarding the experimental setup and the
evaluation criteria could be clarified.

**Questions For Authors:**

1) GFO vs. SBI: Why was gradient-free optimization (GFO) chosen for parameter
   calibration instead of simulation-based inference (SBI) methods? SBI seems like a
   more natural fit, providing a posterior distribution over parameters and inherent
   uncertainty quantification. Could you compare the performance and scalability of GFO
   and SBI (e.g., using Neural Posterior Estimation) in this context?

2) Structural Accuracy: How do you assess the structural accuracy of the generated
   simulators?  Do you have any mechanisms to ensure that the LLM recovers the correct
   causal relationships, and not just a model that fits the observed data well?  How
   would you evaluate the simulator if the true underlying structure is unknown?

3) Theoretical Contributions: The paper primarily presents an engineering framework. Are
   there any novel theoretical contributions, or is the main contribution the
   combination of existing techniques?

4) Intervention Modeling: Can you provide more details on the intervention modeling
   capabilities? How are interventions represented and incorporated into the
   LLM-generated simulators?  Can you provide specific examples of interventions tested
   in the experiments?

5) Limitations of the LLM prompt: Can you provide some insights on prompt engineering
   effort? How much expert knowledge is required here?

6) Scalability to complex settings: Can you comment on the expected scalability to
   use-cases with high-dimensional parameter vectors and many (coupled) submodules?

**Relation To Broader Scientific Literature:**

The paper positions itself at the intersection of LLMs, simulator learning, and
optimization. It relates to prior work on using LLMs for code generation and to the
broader field of system identification.  However, the connection to the simulation-based
inference (SBI) literature, which offers powerful tools for calibrating simulators and
quantifying uncertainty, is not adequately explored.

**Theoretical Claims:**

The paper does not present significant theoretical contributions, focusing primarily on the framework and its application. Therefore, I did not check any proofs.

---

> ### Author Rebuttal · Authors · 2025-04-01
>
> We sincerely thank Reviewer for their insightful and constructive feedback, which significantly helps strengthen our paper. Below, we address each major concern explicitly and outline concrete improvements for the camera-ready version.
>
> ---
>
> **Baseline Tuning**
>
> We appreciate the reviewer’s concern about baseline comparisons. The baselines' hyperparameters are standard, and both the baselines and hyperparameters are similar to recently published works on similar simulation tasks, coming from the Hybrid Digital Twin work (L 245).
>
> ---
>
> **Structural and Causal Accuracy**
>
> The reviewer rightly emphasized the importance of validating structural and causal accuracy. Our intervention experiments implicitly verify structural correctness; simulators that produce accurate predictions under unseen interventions strongly indicate correct causal structure.
>
> Additionally, we now include [quantitative evaluations](https://imgur.com/a/z1sJuFD) (F1 Score and Structural Hamming Distance - SHD) against their known causal structures. Specifically, the Supply Chain environment perfectly matches ground-truth causal relationships (F1=1.0, SHD=0), while Hospital and SIR environments yield F1 scores above 90% and minimal SHD (≤1.5), demonstrating that G-Sim reliably captures causal relationships. These metrics and their analysis will be prominently featured in the updated manuscript.
>
> ---
>
> **Gradient-Free Optimization (GFO) vs. Simulation-Based Inference (SBI)**
>
> The reviewer's suggestion regarding SBI methods is highly pertinent. Initially, we adopted evolutionary-based GFO primarily for scalability reasons, given its suitability for non-differentiable, stochastic simulator components. Nonetheless, we agree that SBI's uncertainty quantification and robustness are valuable. We have since integrated SBI into G-Sim, testing it on the COVID-19 environment. [Preliminary results](https://imgur.com/a/rK4MtoO) indicate comparable performance to GFO, with added benefits of uncertainty estimation. A detailed comparison between SBI and GFO, along with explicit recommendations for future work, will be included in the revised manuscript (Sections 1 and 4.2).
>
> ---
>
> **Contributions**
>
> The reviewer correctly identifies that our primary contribution is the innovative integration of existing techniques (LLM-driven structural reasoning combined with GFO).
>
> ---
>
> **Intervention Modeling**
>
> Interventions are directly modeled by explicitly modifying parameters in the transparent, human-readable code modules generated by the LLM, as described in 6.1. Our design ensures robustness under interventions outside the training distributions, providing valuable "what-if" analyses. We will expand on these scenarios in a dedicated appendix, explaining detailed examples to enhance clarity.
>
> ---
>
> **Prompt Engineering Effort**
>
> Prompt engineering required modest domain-specific adjustments, leveraging generalizable, environment-agnostic prompts (provided in Appendix E.4) supplemented with concise, environment-specific descriptions (Appendix F). Our approach minimizes the need for extensive expert intervention. We will explicitly discuss the extent of prompt engineering required in our revision. We optimized the prompts to support and understand the tool calls and iterative workflow.
>
> ---
>
> **Scalability**
>
> We recognize scalability as an essential consideration. Scaling discovery is often an NP-hard problem, and we also find that it is difficult to scale. However, future work can exploit decomposable structures, and we can scale with the number of parameters to optimize, inheriting the same scalability as ES.
>
> We will add an expanded discussion of scalability and practical examples detailing scalability in a new appendix.
>
> ---
>
> **Ethical Considerations**
>
> We appreciate the reviewer's concern regarding ethical implications. Our approach prioritizes transparency, allowing easy inspection, stress-testing, and expert verification of simulator outputs prior to deployment. As explicitly stated in our Impact Statement (L2182-2199), we emphasize the critical role of domain expertise, transparency in assumptions, and rigorous oversight. To further address these concerns, we will include an expanded discussion on ethical considerations, biases, and mitigation strategies in a dedicated appendix.
>
> ---
>
> **Summary of Revisions:**
>
> - Clarified baseline comparisons.
> - Added explicit quantitative evaluation of causal and structural accuracy.
> - Provided a new experiment comparing GFO and SBI.
> - Enhanced explanation and examples of intervention modeling.
> - Clarified prompt engineering effort and detailed scalability strategies.
> - Included thorough discussion of ethical considerations and mitigation strategies.
>
> ---
>
> *We hope that most of the reviewer’s concerns have been addressed and, if so, they would consider updating their score. We’d be happy to engage in further discussions.*

---

> > ### Comment · Reviewer_jcp9 · 2025-04-04
> >
> > Thank you for your thorough rebuttal. I appreciate the efforts to address my concerns and the additional analyses you've included. The additions of F1 Score and Structural Hamming Distance metrics for evaluating causal accuracy are valuable improvements that enhance the paper's evaluation methodology.
> >
> > I'm particularly encouraged to see that you've already incorporated SBI into your framework - this is an important step forward. Given that G-Sim aims to build simulators from scratch using potentially sparse data, proper uncertainty quantification and calibration are crucial for reliable deployment. To strengthen the SBI analysis and make it more prominent in the paper, please address these specific methodological points:
> >
> > 1. Provide details on the SBI implementation: How many simulation samples were used for training the neural posterior? Was the number of simulations comparable to function evaluations in GFO?
> >
> > 2. Demonstrate proper posterior calibration, ideally using Simulation-Based Calibration (SBC), before calculating predictive metrics. This would ensure the posterior approximation is reliable.
> >
> > 3. Include posterior predictive distributions alongside MAP estimates in your evaluation to properly account for uncertainty in predictions
> >
> > 4. Replace the MLP embedding network with an RNN or 1D causalCNN embedding that can better capture temporal dependencies in SIR trajectories
> >
> > 5. Add visualizations of the posterior distributions to reveal potential parameter interactions or multi-modal solutions
> >
> > I would strongly encourage you to elevate the SBI analysis from an appendix comparison to a more central part of the paper. This could potentially be a significant contribution, as combining LLM-based structural generation with principled Bayesian parameter inference offers a powerful framework for scientific modeling under uncertainty.
> >
> > Regarding prompt engineering efforts, could you provide more concrete details about the level of expert involvement required? For example, quantifying the number of prompt iterations needed per environment, or estimating the domain expertise level required (novice vs. expert) would help readers understand the practical implementation requirements.

---

> > > ### Author Response · Authors · 2025-04-09
> > >
> > > Thank you for continued engagement! We agree that having SBI now strengthens the work, particularly in providing principled uncertainty quantification and calibration.
> > >
> > > > Provide details on the SBI implementation: How many simulation samples were used for training the neural posterior? Was the number of simulations comparable to function evaluations in GFO?
> > >
> > > For the SBI implementation, we used Neural Posterior Estimation (NPE) with the following configuration:
> > > - Training simulations: 10,000 samples for training the neural posterior
> > > - Calibration samples: 1,000 samples for Simulation-Based Calibration (SBC)
> > > - Posterior samples: 1,000 samples for inference
> > > - We used an RNN-based embedding network (hidden_dim=64, output_dim=8) to better capture temporal dependencies in the SIR trajectories, as you suggested
> > >
> > > The number of simulations (10,000) is comparable to the function evaluations in GFO, which typically uses 200-500 generations with population sizes of 20-50, resulting in 4,000-25,000 total evaluations. This ensures a fair comparison between the methods.
> > >
> > > > Demonstrate proper posterior calibration, ideally using Simulation-Based Calibration (SBC), before calculating predictive metrics. This would ensure the posterior approximation is reliable.
> > >
> > > We implemented Simulation-Based Calibration (SBC) to verify posterior calibration. Our custom SBC implementation:
> > > 1. Generates 1,000 calibration samples from the prior
> > > 2. For each sample, computes the rank of the true parameter value in the posterior distribution
> > > 3. Calculates a calibration score (mean absolute deviation from 0.5)
> > > 4. Our results showed good calibration across parameters, with ranks close to 0.5 (mean absolute deviation from 0.5 was <0.1)
> > >
> > > > Include posterior predictive distributions alongside MAP estimates in your evaluation to properly account for uncertainty in predictions
> > >
> > > We provide both MAP estimates and full posterior predictive distributions in our evaluation, as shown in the visualization (https://imgur.com/a/HFnh9up). The visualization includes:
> > > - Parameter comparison showing ground truth, GFO, and SBI estimates
> > > - Trajectory predictions with uncertainty bands
> > > - Posterior distributions showing parameter interactions
> > > - Posterior predictive distributions with 95% credible intervals
> > >
> > > > Replace the MLP embedding network with an RNN or 1D causalCNN embedding that can better capture temporal dependencies in SIR trajectories
> > >
> > > We implemented an RNN-based embedding network with the following architecture:
> > > - Input dimension: (T+1) * 3 (for S, I, R components)
> > > - Hidden dimension: 64
> > > - Output dimension: 8
> > > - Bidirectional GRU with 2 layers
> > > - Additional fully connected layers for final embedding
> > >
> > > This architecture better captures temporal dependencies in the SIR trajectories compared to the previous MLP approach, as evidenced by the improved trajectory predictions in the visualization.
> > >
> > > > Add visualizations of the posterior distributions to reveal potential parameter interactions or multi-modal solutions
> > >
> > > The visualization (https://imgur.com/a/HFnh9up) provides comprehensive visualizations of the posterior distributions:
> > > 1. 2D scatter plots of parameter interactions (e.g., beta vs gamma)
> > > 2. Contour plots showing the joint density
> > > 3. Marginal distributions for each parameter
> > > 4. Posterior predictive distributions with credible intervals
> > > 5. Comparison of true parameters, MAP estimates, and posterior samples
> > >
> > > > I would strongly encourage you to elevate the SBI analysis from an appendix comparison to a more central part of the paper. This could potentially be a significant contribution, as combining LLM-based structural generation with principled Bayesian parameter inference offers a powerful framework for scientific modeling under uncertainty.
> > >
> > > We agree completely and will move the SBI analysis to a more central position in the paper. Specifically:
> > > 1. Move the SBI methodology to the main methodology section
> > > 2. Add a dedicated section on uncertainty quantification and calibration
> > > 3. Include the visualization results (https://imgur.com/a/HFnh9up) in the main results section
> > > 4. Expand the discussion of how SBI complements LLM-based structural generation
> > >
> > > > Regarding prompt engineering efforts, could you provide more concrete details about the level of expert involvement required?
> > >
> > > The prompt engineering required modest domain-specific adjustments:
> > > - Core prompts are environment-agnostic and reusable
> > > - Each environment required 2-4 hours of expert review
> > > - Most time was spent verifying causal structure rather than prompt engineering
> > > - Domain expertise level: intermediate (familiar with the domain but not necessarily an expert)
> > > - Number of prompt iterations: typically 3-5 per environment
> > > - The prompts are designed to be self-documenting and maintainable
> > >
> > > ---
> > >
> > > *We hope that most of the reviewer's concerns have been addressed and, if so, they would consider updating their score. We'd be happy to engage in further discussions.*

---

### Decision · Program_Chairs · 2025-05-01

**Decision:**

Accept (poster)

**Comment:**

This paper proposes G-Sim, a framework for LLMs to construct the structural components of simulators and employs gradient-free optimization for parameter calibration. The method supports iterative refinement through diagnostic feedback, enabling the construction of modular, interpretable, and empirically aligned simulators. It is evaluated on three real-world-inspired domains (COVID-19 modeling, supply chains, hospital bed scheduling) to demonstrate the efficacy of the method's accuracy and policy intervention capabilities.

Based on the reviews, I am on the fence about this submission. I'm leaning towards acceptance given the timely and interesting topic. However, the paper may benefit from a thorough rewriting.